# Many-Shot CoT-ICL: Making In-Context Learning Truly Learn

**Tsz Ting Chung** [1]  **Lemao Liu** [2]  **Mo Yu** [3]  **Dit-Yan Yeung** [1]

## Abstract

While many-shot ICL achieves remarkable performance, prior studies of its scaling behavior have mainly focused on non-reasoning tasks. In this work, we study many-shot ICL on reasoning tasks, with a particular focus on many-shot chain-of-thought in-context learning (CoT-ICL). Analyzing across non-reasoning and reasoning tasks and across non-reasoning and reasoning-oriented LLMs, we identify several distinctive properties of many-shot CoT-ICL. We further interpret these findings by viewing many-shot CoT-ICL as in-context test-time learning rather than scaled pattern matching, and suggest two principles: (i) demonstrations should be easy for the target model to understand, and (ii) they should be ordered to support a smooth conceptual progression. Guided by the principle, we propose Curvilinear Demonstration Selection (CDS), a simple ordering method that yields up to a 5.42 percentage-point gain on a math task with 64 demonstrations. Overall, our results reframe the long context window from a retrieval buffer into a structured curriculum for in-context test-time learning.

## 1. Introduction

In-context learning (ICL) enables large language models (LLMs) to perform tasks by conditioning on a sequence of input-output demonstrations without updating their parameters (Min et al., 2022; Von Oswald et al., 2023). Research has focused on improving ICL through strategies like selecting effective demonstrations (Sorensen et al., 2022; Liu et al., 2022; Wu et al., 2023). Recently, with the expansion of context windows, many-shot ICL has emerged, where dozens to hundreds of demonstrations can be provided,

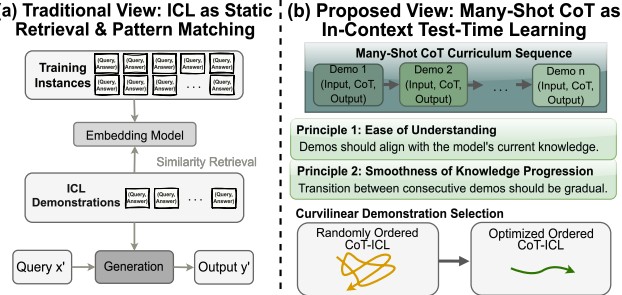

*Figure 1.* Reframing of CoT-ICL as in-context test-time learning.

achieving performance competitive with fine-tuning (Agarwal et al., 2024; Bertsch et al., 2025).

In parallel, chain-of-thought (CoT) prompting has become a standard practice for complex reasoning (e.g., arithmetic and narrative reasoning), where models generate intermediate steps before an answer (Wei et al., 2022; Kojima et al., 2022). At the same time, test-time scaling studies how to improve model performance through additional computation during inference without parameter updates (Snell et al., 2025b; Lin et al., 2024). These threads naturally intersect: many-shot CoT-ICL is a basic form of test-time computation, where long sequences of reasoning demonstrations shape the model's behavior at inference time.

However, there exists a critical gap. Current understanding of many-shot dynamics derives almost entirely from studies of non-reasoning tasks. For example, a consistent finding on many-shot ICL is that for non-reasoning tasks, the impact of demonstration order diminishes with scale (Bertsch et al., 2025; Baek et al., 2025). It remains unknown whether the established findings on many-shot ICL extend to many-shot CoT-ICL for reasoning. Does providing more reasoning demonstrations lead to reliable improvement, or does it introduce new instabilities? These questions are practically important for deploying reasoning-capable LLMs and theoretically fundamental: it probes whether many-shot CoT-ICL for reasoning is merely large-scale pattern matching or a form of genuine learning in in-context learning that follows pedagogical principles.

In this work, we demonstrate that the established rules of many-shot ICL break down for reasoning tasks. Through systematic experiments across model types (non-reasoning vs. reasoning-oriented) and tasks (non-reasoning vs. reason-

---

[1]Hong Kong University of Science and Technology [2]Fudan University [3]Wechat AI, Tencent. Correspondence to: Lemao Liu <lemaoliu@fudan.edu.cn>, Dit-Yan Yeung <dyyeung@ust.hk>.

*Proceedings of the 43rd International Conference on Machine Learning*, Seoul, South Korea. PMLR 306, 2026. Copyright 2026 by the author(s).

ing), our experiments uncover: (1) a setting-dependent scaling effect, where many-shot ICL scales on non-reasoning tasks but many-shot CoT-ICL on reasoning tasks is unstable for non-reasoning LLMs and improves mainly for reasoning-oriented LLMs; (2) that similarity-based retrieval explains non-reasoning scaling but fails on reasoning because question similarity does not ensure procedural compatibility, pointing to in-context learning beyond surface matching; and (3) an order-scaling effect, where performance variance grows with the number of CoT demonstrations.

We explain these results by reframing many-shot CoT-ICL as in-context test-time learning rather than pattern matching. We propose that successful demonstrations must be both understandable to the model and smoothly sequenced. We examine this through two principles: (1) The Ease of Understanding: demonstrations should align with the model's current knowledge (explaining why self-generated demonstrations work best for weaker models); and (2) The Smoothness of Knowledge Progression: the conceptual transition between consecutive demonstrations should be gradual (quantifiable via the curvature of their embedding trajectory) as illustrated in Figure 1. Building on these principles, we introduce Curvilinear Demonstration Selection (CDS), a practical method that orders demonstrations to minimize total conceptual curvature. This approach yields an average of 3.81% performance gains across both math and narrative reasoning tasks.

Our contributions are threefold: (1) We characterize the scaling dynamics of many-shot CoT-ICL; (2) We reframe effective many-shot CoT through the ease of understanding and smoothness of information flow, bridging ICL with insights from test-time learning; (3) We introduce and validate a practical, principle-driven method for demonstration ordering that advances many-shot reasoning.

## 2. Related Works

**Many-shot ICL**  The extension of LLM context windows (Peng et al., 2024; Han et al., 2024) has enabled many-shot ICL, where models process significantly more demonstrations (Agarwal et al., 2024; Bertsch et al., 2025; Chung et al., 2024). Wan et al. (2025) show that many-shot gains are often driven by a subset of influential examples, and that similarity-based retrieval is ineffective compared with validation-guided optimization and generation. We instead study when many-shot CoT-ICL is effective: when scaling helps across task and model types, why semantic similarity fails for reasoning, and how demonstration order shapes in-context procedural learning. Findings in broader many-shot ICL reveal that with sufficient demonstrations, model sensitivity to ordering diminishes for standard classification tasks (Baek et al., 2025; Bertsch et al., 2025), suggesting a form of robustness with scaling. This led to a narrative

that in many-shot settings, careful demonstration engineering may be unnecessary. However, these studies focused overwhelmingly on non-reasoning tasks (e.g., classification, simple QA) (Baek et al., 2025; Bertsch et al., 2025), neglecting performance on reasoning tasks (Hendrycks et al., 2021; Chung et al., 2025; Xu et al., 2024; Yu et al., 2025a). Concurrent work on test-time scaling leverages extended computation for self-improvement without parameter updates (Snell et al., 2025b; Li et al., 2025b), suggesting that effective in-context learning can be viewed as a form of real-time optimization. Our work connects many-shot CoT-ICL to test-time learning, guided by two key principles that explain how learning occurs inside the context.

**Chain-of-Thought**  CoT prompting (Wei et al., 2022) decomposes reasoning into intermediate steps, substantially improving LLM performance on complex tasks. Subsequent studies like Tree-of-Thoughts (Yao et al., 2023) and Program-of-Thoughts (Chen et al., 2023) explore structured reasoning paths, while methods like rStar-Math (Guan et al., 2025) employ search algorithms for trajectory optimization. These approaches primarily focus on enhancing the reasoning process for a single query. In the ICL setting, Dr.ICL (Luo et al., 2023) demonstrates that retrieving relevant CoT demonstrations boosts few-shot performance, and Auto-CoT (Zhang et al., 2023) proposes an automatic few-shot CoT prompting method that clusters questions to sample diverse representatives and generate reasoning chains as demonstrations. However, a critical gap remains: existing CoT-ICL work largely operates in few-shot settings. The fundamental question of how CoT demonstrations scale with context length and whether the principles of effective demonstration design change from few-shot to many-shot is largely unexplored. Our work positions many-shot CoT not merely as "more examples", but as a potential in-context curriculum that requires principled sequencing.

**Demonstration Selection**  Demonstration selection has long been studied for effective few-shot ICL. The dominant paradigm is similarity-based retrieval, where demonstrations semantically closest to the test query are selected (Liu et al., 2022; Wu et al., 2023; Kapuriya et al., 2025). This approach implicitly frames ICL as a form of pattern matching (Olsson et al., 2022; Crosbie & Shutova, 2025; Yu et al., 2025b). Interestingly, this paradigm finds a direct analogy in Retrieval-Augmented Generation (RAG), where relevant context chunks are retrieved via embedding similarity (Lewis et al., 2020). Our work challenges whether this conclusion extends to reasoning tasks. We hypothesize that for CoT-ICL, effective demonstration selection is less about retrieving semantically similar examples and more about constructing a smooth learning sequence that facilitates conceptual understanding, acting as a shift from "retrieval for matching" to "retrieval for learning".

# 3. Settings

We establish an experimental framework for studying many-shot In-Context Learning (ICL), with and without Chain-of-Thought (CoT), under long-context constraints. Our design spans three dimensions: *task type* (non-reasoning vs. reasoning), *model type* (standard instruction-tuned vs. explicitly "reasoning" models), and *ICL configuration* (prompt format and number of demonstrations).

## 3.1. Tasks Studied

Prior many-shot work has largely emphasized non-reasoning classification benchmarks (Li et al., 2025a; Bertsch et al., 2025). We extend evaluation to include both classification-style tasks and multi-step reasoning tasks, while using a unified *open-ended generation* evaluation for all datasets.

**Evaluation protocol.** For each test instance, the model generates a free-form text completion. We map the completion to a predicted answer using task-specific extraction and normalization, and score it by *exact match* against the reference. Prompt templates for evaluation are provided in Appendix E. For numerical datasets (e.g., GSM8K/MATH), we extract the final numeric value or mathematical expression from the completion and compare it to the ground truth under the same exact-match criterion.

**Non-reasoning tasks.** These tasks require little intermediate reasoning and primarily test semantic understanding and label mapping. We include benchmarks with different label-space sizes: SuperGLUE (Wang et al., 2019) (small label space), NLU (nlu, 2021), TREC (Hovy et al., 2001), and BANKING77 (Casanueva et al., 2020) (large label space).

**Reasoning tasks.** These tasks require deduction and/or mathematical derivation. We focus on mathematical reasoning with GSM8K (Cobbe et al., 2021) and MATH (Hendrycks et al., 2021), and include DetectiveQA (Xu et al., 2024) for narrative reasoning over long contexts. For tasks that provide gold rationales, we use the dataset-provided reasoning chains $C_i$ as the CoT component in demonstrations (Section 3.3).

## 3.2. LLMs Studied

We evaluate a range of LLMs and group them by whether they explicitly contain extended reasoning at inference time.

**Non-reasoning LLMs.** These models are primarily tuned to produce direct answers given instructions, without an explicit "thinking" token. We evaluate LLaMA 3.1 (`Llama-3.1-8B-Instruct`), LLaMA 3.3 (`Llama-3.3-70B-Instruct`) (MetaAI, 2024), Qwen 2.5 (7B) (`Qwen2.5-7B-Instruct`), and Qwen 2.5 (14B)

(`Qwen2.5-14B-Instruct`).

**Reasoning-oriented LLMs.** These models expose an explicit reasoning segment (e.g., a `<think>` token). We evaluate Qwen 3 (8B) (`Qwen3-8B`) and Qwen 3 (14B) (`Qwen3-14B`) (Yang et al., 2025), QwQ (32B) (`QwQ-32B`) (Qwen et al., 2024), and DeepSeek-R1 (685B) (`DeepSeek-R1`) (DeepSeek-AI, 2025). For reasoning-oriented models, we enable the model's reasoning mode during inference to allow the generation of intermediate reasoning tokens.

**Long-context configuration.** To support many-shot prompts (up to 131K tokens for Qwen-family models), we apply the official RoPE scaling configurations provided by each model release. All other decoding and system-prompt settings follow the model providers' recommended defaults unless stated otherwise.

## 3.3. ICL Configuration

We study scaling from few-shot to many-shot under two prompting paradigms.

**Traditional ICL.** Prompts consist of $k$ input–output pairs $(x_i, y_i)$ followed by a query $x'$. The model produces an output $y'$ conditioned on the ordered demonstration sequence:

$$y' = \text{LLM}(x' \mid [(x_1, y_1), \dots, (x_n, y_n)]). \quad (1)$$

**CoT-ICL.** Prompts consist of $k$ triples $(x_i, C_i, y_i)$, where $C_i$ is a reasoning chain. Given a query $x'$, the model generates both an intermediate chain $C'$ and a final answer:

$$(C', y') = \text{LLM}(x' \mid [(x_1, C_1, y_1), \dots, (x_n, C_n, y_n)]). \quad (2)$$

**Context scaling.** CoT demonstrations are substantially longer than standard ICL examples (e.g., in our geometry setting, a single CoT demonstration can be $\sim 30\times$ longer than a BANKING77 example). As a result, while hundreds to thousands of demonstrations may fit for traditional ICL, CoT-ICL is typically limited to at most a few hundred demonstrations by context length. We therefore focus our scaling analysis on $n \leq 128$, which captures the most informative trade-offs between model type, task type, and demonstration count in our long-context regime.

# 4. Properties of CoT-ICL

## 4.1. Scaling with Reasoning Tasks

Prior work reports that many-shot ICL yields reliable improvements on non-reasoning tasks (Bertsch et al., 2025; Baek et al., 2025). We replicate this behavior, but find

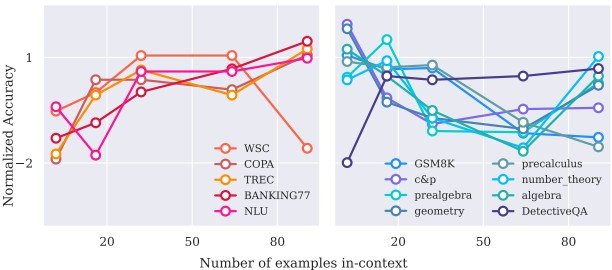

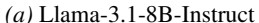

*(a)* Llama-3.1-8B-Instruct

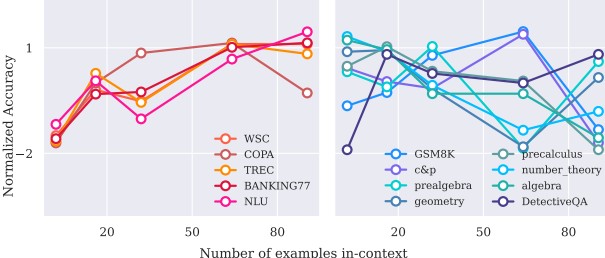

*(b)* Qwen2.5-7B-Instruct

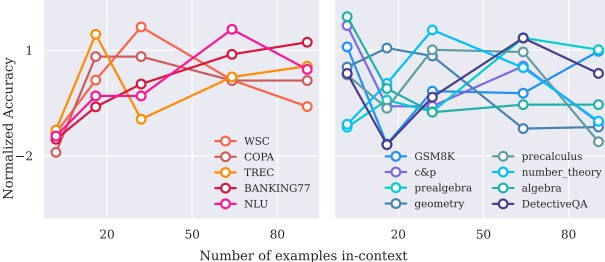

*(c)* Qwen2.5-14B-Instruct

*Figure 2.* Scaling disparity between task types. Performance (normalized accuracy) of non-reasoning LLMs on classification tasks (warm colors) versus reasoning tasks (cool colors). The x-axis represents normalized accuracy (i.e., $\frac{x - \bar{x}}{\sigma_x}$ for accuracy $x$), while the y-axis indicates the number of in-context demonstrations.

it does *not* extend to reasoning tasks when demonstrations include CoT rationales. Figure 2 shows a clear contrast: non-reasoning tasks improve steadily as the number of demonstrations increases, whereas reasoning performance is unstable and often degrades for non-reasoning LLMs.

This failure is not explained by insufficient parameter scale. As shown in Figure 3 (left), even Llama 3.3 70B can incur negative gains from adding more CoT demonstrations. Together, these results suggest a qualitative difference between scaling traditional ICL and CoT-ICL.

### 4.2. Scaling with Reasoning LLMs

The scaling behavior changes markedly for models with explicit reasoning capabilities. Figure 3 (right) shows that QwQ (32B) and R1 (685B) improve consistently as more CoT demonstrations are added. This trend also holds for

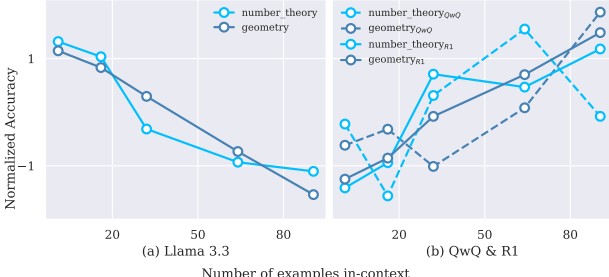

(a) Llama 3.3      (b) QwQ & R1

Number of examples in-context

*Figure 3.* Scaling disparity between model types on math reasoning tasks. *Left:* Llama 3.3 (non-reasoning LLM) shows negative gains. *Right:* QwQ (32B) and R1 (685 B) (reasoning LLM) shows clear positive scaling.

| Model | geometry | number_theory | DetectiveQA |
|---|---|---|---|
| Qwen3-14B (en) | **73.07** | **91.30** | 72.73 |
| Qwen3-14B (dis) | 65.76 | 88.15 | 72.73 |
| Qwen3-8B (en) | **67.01** | **84.63** | **69.48** |
| Qwen3-8B (dis) | 62.63 | 79.81 | 66.88 |

*Table 1.* Performance at $n = 128$ with reasoning-oriented models' thinking mode (**en**)abled versus (**dis**)abled.

smaller reasoning-optimized models: across the Qwen3 family (Figure 4), performance increases near-monotonically with additional demonstrations.

The divergence between model classes indicates that benefiting from long CoT contexts is not a generic consequence of having more examples in context. Instead, positive scaling appears to require model mechanisms that can use demonstrations as intermediate reasoning signal (e.g., via thinking tokens and/or reasoning-oriented training), rather than relying primarily on shallow pattern matching. To directly test this interpretation, we evaluate the same $n = 128$ many-shot CoT contexts with thinking enabled versus disabled. As shown in Table 1, suppressing the generation of intermediate reasoning hurts performance on geometry and number_theory for both Qwen3 models, and also hurts DetectiveQA for Qwen3-8B. Furthermore, when thinking is enabled on geometry, increasing $n$ from 16 to 128 improves Qwen3-14B accuracy from 66.18% to 73.07%, while reducing the average number of generated tokens inside the `<think>` segment by 24.02%. This suggests that larger CoT contexts help the model internalize task procedures, reducing the need for verbose query-time deliberation.

### 4.3. Rethinking ICL with similarity

Sections 4.1–4.2 reveal a consistent split: many-shot ICL scales reliably on non-reasoning tasks, while many-shot CoT-ICL for reasoning is unstable for non-reasoning LLMs and improves mainly for reasoning-optimized LLMs.

For positive scaling effect, a common explanation for why many-shot ICL works is the *retrieval hypothesis*: additional demonstrations help because the model can locate and reuse

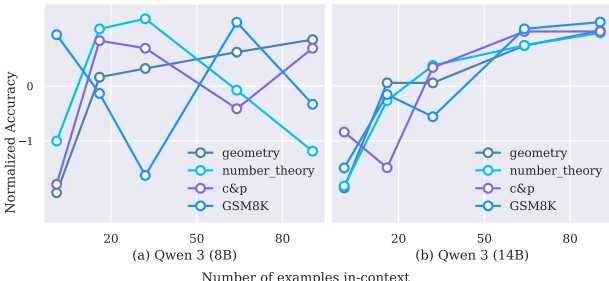

*Figure 4.* Positive scaling of reasoning LLMs. The Qwen3 family (reasoning LLMs) demonstrates consistent performance improvements with more demonstrations on math reasoning tasks. *Left:* Qwen3 (8B) *Right:* Qwen3 (14B)

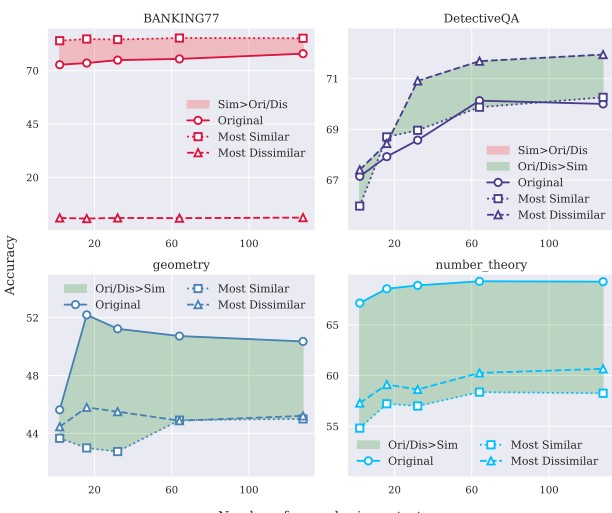

*Figure 5.* Performance with original(ori), similarity(sim) and dissimilar(dis) sets averaged across five LLMs. The area between the two sets is filled with colors, indicating the relative performance.

examples that are semantically similar to the query (Liu et al., 2022; Wu et al., 2023). If many-shot CoT-ICL for reasoning were driven by the same mechanism, then (i) retrieving question-similar demonstrations should help more as $k$ grows, and (ii) the most-similar set should consistently outperform dissimilar or uncurated sets.

For each test query, we embed all candidate *training questions* (question-only) with Qwen3-Embedding-4B (Zhang et al., 2025) and rank candidates by cosine similarity. We then build two $k$-shot demonstration sets per query: (i) *most-similar* (top-$k$) and (ii) *most-dissimilar* (bottom-$k$), keeping the original CoT+answer paired with each selected question. We evaluate five base LLMs (Llama 3.1, Qwen 2.5 7B/14B, Qwen3 8B/14B) and report averages; details are in Appendix A.

*Similarity retrieval succeeds for non-reasoning tasks, but fails for reasoning tasks.* Figure 5 supports the retrieval hypothesis on a non-reasoning task BANKING77. The most-similar sets consistently outperform the most-dissimilar sets. However, the same heuristic breaks on reasoning tasks. Across geometry, number_theory, and DetectiveQA, the most-similar sets are consistently *worse* than either the most-dissimilar sets or the original (unretrieved) sets. This conclusion holds when evaluating reasoning and non-reasoning LLMs separately (Appendix A.5).

*Similarity optimizes matching, not learning.* These results align with the paper's central message: many-shot CoT-ICL for reasoning is not well explained as scaled-up pattern matching. For non-reasoning tasks, question-level similarity is often a reliable proxy for label similarity, so retrieving similar demonstrations improves performance. For reasoning tasks, in contrast, question-level similarity is a weak proxy for procedural compatibility. Two problems can look semantically similar while requiring different solution strategies, and their associated CoT-ICL may induce conflicting intermediate steps. We provide qualitative examples and additional analysis in Appendix A.4.

This provides a mechanism-level explanation for the negative scaling observed in Section 4.1. Solving reasoning tasks depends on extracting and reusing *procedures*, not merely matching surface patterns. With purely surface matching, LLMs are likely to be misled by a set of "similar" but procedurally mismatched CoTs, leading to negative gains with similar retrieval.

The failure of similarity-based retrieval with reasoning LLMs also suggests that the mechanism behind positive scaling differs across settings. In particular, the rationale for why scaling works for reasoning-oriented LLMs on reasoning tasks (Section 4.2) is not the same as why scaling works for non-reasoning LLMs on non-reasoning tasks (Section 4.1). From a learning perspective, a plausible explanation is that reasoning-oriented models can better interpret the provided CoT-ICL and extract higher-level procedural structure in the thinking content beyond surface pattern matching, allowing the benefit from additional demonstrations.

### 4.4. Ordering Stability of CoT-ICL

If CoT demonstrations act as a learning signal rather than a static reference, their *order* should matter, since order changes the trajectory of intermediate states induced by the context. This prediction contrasts with findings on non-reasoning tasks, where order sensitivity decreases as the number of demonstrations grows (Bertsch et al., 2025; Baek et al., 2025).

We quantify order sensitivity by sampling five random permutations of the same demonstration set and measuring the standard deviation of accuracy. For non-reasoning tasks, we reproduce the low-variance behavior (Figure 6, left). In con-

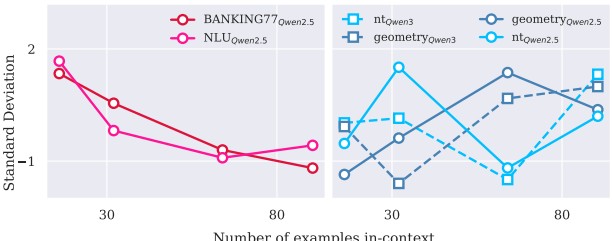

*Figure 6.* Standard deviation of performance across five random demonstration orders on classification tasks (warm colors) versus reasoning tasks (cool colors), where nt corresponds to number_theory. Results shown for Qwen2.5 (14B) (non-reasoning) and Qwen3 (14B) (reasoning).

trast, for reasoning tasks we observe the opposite trend: variance *increases* with more demonstrations (Figure 6, right). This holds for both non-reasoning and reasoning LLMs.

Overall, many-shot CoT-ICL exhibits strong and growing path dependence: performance depends not only on *which* demonstrations are provided, but also on *how* they are sequenced. This instability is consistent with CoT-ICL behaving as an in-context learning process whose effectiveness depends on the induced reasoning trajectory, motivating our in-context test-time learning perspective in the next section. We further validate these conclusions by computing mean and standard deviation across five random demonstration-ordering seeds on an independently sampled ICL subset. The same qualitative trends persist for reasoning-oriented models, non-reasoning models, and cross-model CoT transfer, with full results in Appendix B.

# 5. Rethinking ICL: From Pattern Matching to In-Context Test-Time Learning

Sections 4.4 and 4.3 suggest that many-shot CoT-ICL does not behave like a simple nearest-neighbor or pattern-matching mechanism: increasing the number of demonstrations amplifies order sensitivity, and similarity-based selection is not reliably helpful on reasoning tasks. We therefore adopt a different lens: *in-context test-time learning*, where the prompt serves as training data and the forward pass resembles a gradient-free form of adaptation. Under this view, demonstrations do not only provide *answers to copy*; they can shape an internal procedure for how to solve the task.

Before deriving design principles from this view, we first provide direct evidence that models indeed absorb procedures from demonstrations rather than merely exploiting input–output associations.

**Direct evidence for procedure absorption.** We further test whether the model uses demonstration-specific procedures rather than only the input–output mapping. On geometry, we compare standard many-shot CoT demonstrations,

| Model | Setting | $n = 16$ | $n = 128$ |
|---|---|---|---|
| Qwen3-8B | Valid | 57.62 | **67.01** |
| | Corrupted | 57.62 | 65.76 |
| Qwen3-14B | Valid | 66.17 | **73.07** |
| | Corrupted | **67.01** | 70.56 |

*Table 2.* Procedural-corruption ablation on geometry. The larger drop at $n = 128$ indicates that models use demonstration-specific procedures, not only answer labels or long-context activation.

$(x_i, C_i, y_i)$, against a procedural-corruption condition that preserves every question and final answer but replaces all rationales with the same static chain from the first demonstration, $(x_i, C_0, y_i)$. This controls for format, context length, and the $x \to y$ mapping, isolating whether the aligned procedure $C_i$ matters. Table 2 shows that at $n = 16$ the two settings are nearly indistinguishable, but at $n = 128$ corrupted procedures cause clear drops for both Qwen3-8B and Qwen3-14B. Thus, when enough informative rationales are provided, reasoning models appear to use the procedural steps in the demonstrations rather than merely memorizing answer labels or passively activating long-context priors.

This framing yields two practical principles for demonstration design. First, demonstrations must be *understandable* to the model (otherwise they cannot be used as supervision at test time). Second, demonstrations should be arranged to yield a *smooth information flow* across the prompt (otherwise the induced procedure becomes unstable), providing a direct explanation for the order sensitivity observed in Section 4.4. These principles also connect to recent evidence that scaling test-time computation improves performance by refining internal solution procedures (Snell et al., 2025a).

## 5.1. Principle 1: Ease of understanding

If ICL operates as in-context test-time learning, then demonstrations must fall within the model's current ability to parse and internalize. In educational psychology, effective instruction targets a learner's "zone of proximal development" (Benson, 2020), the range between what they can solve unaided and what they can solve with appropriate guidance. By analogy, we posit a *zone of understandable reasoning*: demonstrations are most useful when the model can follow their reasoning steps and internalize the implied procedure, rather than when they are merely "higher quality" but stylistically or procedurally misaligned with the model.

### 5.1.1. SETTINGS

We test whether demonstration effectiveness depends more on *answer correctness* or on *alignment with the model's own generation distribution*. For each training instance, we sample CoT demonstrations from each LLM with temperature 1.0 (10 samples per instance) and construct:

- *Correct* (cr): generated CoT with correct answers.
- *Wrong* (wr): generated CoT with incorrect answers.

- *First* (`first`): the first sampled CoT regardless of correctness.

We compare these sets against the dataset-provided ground-truth CoT (`origin`).

We use `cr`/`wr` for the Qwen 2.5 family, where incorrect generations are sufficiently frequent to construct a sizable `wr` set, except for the GSM8K task. For Qwen 3 family, which achieves higher accuracy and therefore rarely produces wrong answer under our sampling budget, constructing `wr` is difficult. We instead use the `first` set to evaluate the effect of self-generated (and distribution-aligned) demonstrations without conditioning on correctness. We additionally evaluate cross-model transfer by using CoTs generated by stronger models (Qwen2.5/3 14B) as demonstrations for weaker models.

### 5.1.2. RESULTS

**(1) Understanding improves with *distributional alignment*: self-generated CoTs perform best.** Figures 7 and 8 show that prompts constructed from self-generated demonstrations (`cr`/`wr`/`first`) consistently outperform dataset-provided CoTs (`origin`) and cross-model demonstrations written by stronger LLMs when used to prompt weaker ones, even when some self-generated demonstrations have incorrect final answers. Because self-generated CoTs are drawn from the target model's own generation distribution, they are more likely to be *understandable* to that model (i.e., easier to condition on and reuse as procedural supervision), consistent with Principle 1.

**(2) Understanding improves with scale: the self-generated advantage diminishes for stronger models.** If demonstration effectiveness is limited by understanding, then as model capability increases it should become easier to extract useful procedures from less-aligned supervision (i.e., `origin`), reducing the relative benefit of self-generated CoTs. Consistent with this intuition, the gain of self-generated demonstrations over `origin` shrinks with model ability (e.g., Qwen3-14B exhibits a smaller gain than Qwen3-8B in Figure 8). This suggests that stronger models can understand and exploit ground-truth rationales more reliably.

**(3) Understanding improves with *reasoning-oriented priors*: reasoning models are less brittle to supervision mismatch.** Beyond scale, Section 4.2 shows that reasoning LLMs outperform non-reasoning LLMs at comparable parameter sizes under the provision of dataset-provided CoT-ICL. A plausible explanation is that the reasoning within the thinking tokens acts as an additional prior that guides how demonstrations are interpreted and how procedural patterns are extracted from them. Under this view, bet-

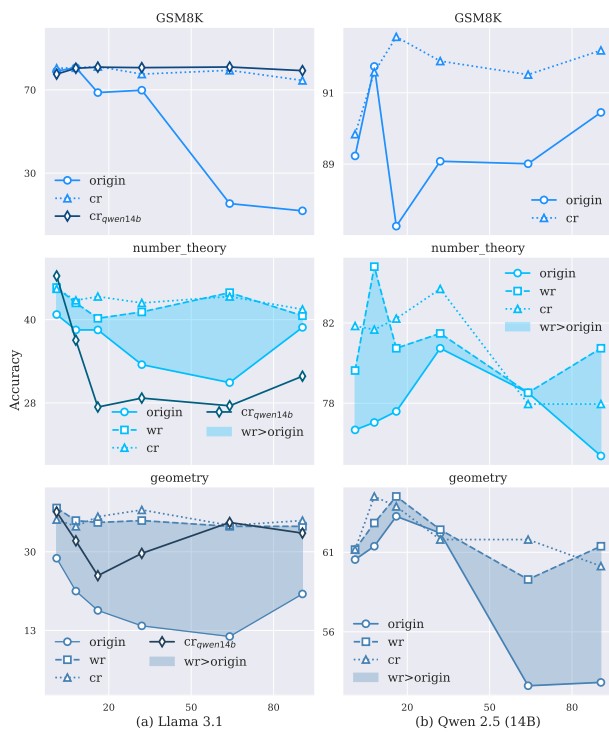

*Figure 7.* Performance of two sets of self-generated in-context CoT, including the set filtered with only correct answer(cr) and the set filtered with only wrong answer(wr). $\text{cr}_{qwen14}$ is prompting the LLaMA model with the in-context CoT generated by Qwen 2.5 (14B). *Left:* Llama 3.1 *Right:* Qwen 2.5 (14B)

ter performance reflects a stronger ability to leverage the supervision signal in provided examples, even when the dataset-provided demonstrations are not perfectly aligned with the target model.

### 5.2. Principle 2: Smoothness of information flow

Effective learning requires not just comprehensible individual examples, but a coherent progression between them. We hypothesize that smooth transitions between demonstrations facilitate the model's construction of a coherent reasoning schema, while abrupt conceptual jumps disrupt this process.

#### 5.2.1. SETTINGS: QUANTIFYING TRANSITION SMOOTHNESS

We measure smoothness by viewing an ordered list of demonstrations as a trajectory in embedding space. We represent each demonstration $\mathbf{d}_i$ as *(question + CoT + final answer)* and embed it using Qwen3-Embedding-4B to obtain $\mathbf{e}_i \in \mathbb{R}^d$. Unlike Section 4.3 (question-only similarity), we embed the full demonstration because ordering effects should depend on procedural content. This representation is designed to capture not only topical similarity but also the logical structures and operations expressed in the CoT rationale. We compute curvature directly in the original embedding space, with details in Appendix C.

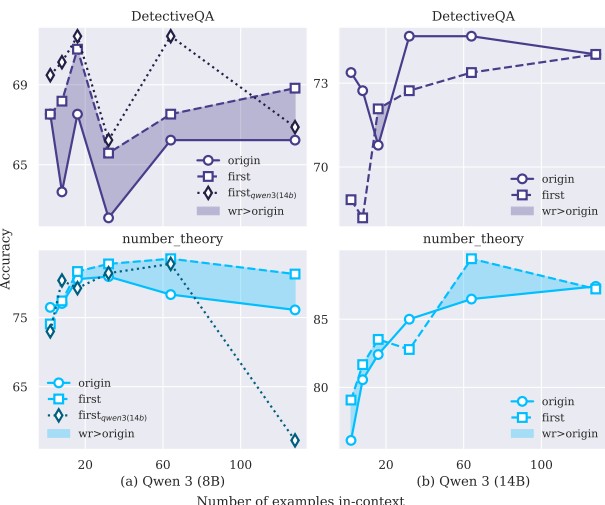

*Figure 8.* Performance of the first set of self-generated in-context CoT. first$_{qwen3(14b)}$ is prompting the Qwen 3 (8B) model with the in-context CoT generated by Qwen 3 (14B). *Left:* Qwen 3 (8B) *Right:* Qwen 3 (14B)

Given an ordering $O = [\mathbf{d}_1, \ldots, \mathbf{d}_n]$, we define local curvature at position $i$ as the turning angle between consecutive displacement vectors:

$$\theta_i = \arccos\left(\frac{(\mathbf{e}_i - \mathbf{e}_{i-1}) \cdot (\mathbf{e}_{i+1} - \mathbf{e}_i)}{\|\mathbf{e}_i - \mathbf{e}_{i-1}\| \, \|\mathbf{e}_{i+1} - \mathbf{e}_i\|}\right) \qquad (3)$$

Total curvature is $\Theta(O) = \sum_{i=2}^{n-1} \theta_i$, where smaller values indicate smoother transitions. Implementation details (including the exact concatenation template and robustness checks) are in Appendix C.

### 5.2.2. RESULTS

Across three math reasoning tasks, ordering curvature is strongly negatively correlated with accuracy: overall $r = -0.547$, with task-wise correlations of $-0.545$ (geometry), $-0.468$ (number_theory), and $-0.628$ (counting_and_probability). Thus, smoother orderings tend to yield better performance.

This also provides a concrete explanation for the increasing order variance observed in Section 4.4. As the number of demonstrations grows, random permutations are more likely to contain sharp "conceptual jumps" (high curvature), amplifying variability across orders. Controlling the ordering to reduce curvature yields a more stable learning trajectory, motivating our ordering method in Section 6.

**Causal smoothness ablation.** To separate smooth transitions from local clustering, we construct two orderings from the same demonstrations using bge-m3 embeddings. Both orderings constrain Euclidean proximity, but the high-curvature baseline inverts the curvature objective, forcing abrupt conceptual turns while preserving local neighbor-

hoods. Across number_theory and geometry, CDS consistently outperforms this high-curvature ordering in Table 4, supporting transition smoothness as a causal factor rather than a by-product of grouping similar examples.

### 5.2.3. DISCUSSION: PEDAGOGICAL ANALOGY

This principle mirrors effective textbook design: concepts are introduced progressively, with each chapter building smoothly upon the previous. Abrupt topic changes or missing prerequisites hinder learning. Similarly, in many-shot CoT-ICL, demonstrations must be ordered to create a "conceptual curriculum" that guides the model from basic to advanced reasoning steps.

## 6. Curvilinear Demonstration Selection

Motivated by the curvature–performance correlation in Section 5.2, we introduce *Curvilinear Demonstration Selection* (CDS), a practical method for constructing an ordering of many-shot CoT demonstrations. CDS aims to produce a smooth trajectory in embedding space, avoiding abrupt transitions between successive demonstrations.

### 6.1. Experiment Settings

We evaluate CDS on three reasoning tasks spanning diverse domains, including geometry, number theory, and DetectiveQA. Our primary experiments use reasoning LLMs from the Qwen3 family (8B and 14B) across multiple demonstration budgets, with the motivation and experimental details provided in Appendix D.1. We additionally include gpt-5.2 in the robustness study to test whether the ordering effect transfers to a closed-source reasoning model.

**Objective.** Given a set of $n$ demonstrations with original embeddings $\{\mathbf{e}_i\}_{i=1}^n$, CDS seeks a permutation $O = [\mathbf{d}_{\pi(1)}, \ldots, \mathbf{d}_{\pi(n)}]$ that minimizes the total curvature

$$\Theta(O) = \sum_{t=2}^{n-1} \arccos\left(\frac{\mathbf{v}_t \cdot \mathbf{v}_{t+1}}{\|\mathbf{v}_t\| \, \|\mathbf{v}_{t+1}\|}\right) \qquad (4)$$

$$\mathbf{v}_t = \mathbf{e}_{\pi(t)} - \mathbf{e}_{\pi(t-1)}. \qquad (5)$$

**TSP-based approximation.** Directly optimizing Eq. (4) is combinatorial for large $n$: exact minimization requires evaluating $n!$ permutations, which is infeasible for the longest prompts studied in this paper ($n \leq 128$). Moreover, optimizing angles alone can produce trajectories that are geometrically straight but make very large jumps across the embedding space. We therefore use a practical TSP-based heuristic with a pairwise transition cost that balances spatial proximity and a local curvature proxy. For each ordered pair

| Task | Model | Method | Number of demonstrations | | | |
|---|---|---|---|---|---|---|
| | | | 16 | 32 | 64 | 128 |
| number_theory | gpt-5.2 | origin | 89.63 | 91.11 | 88.56 | 91.48 |
| | | CDS | 89.26 | **92.59** | **92.04** | **91.85** |
| | | CDS$_{bge}$ | 88.15 | **91.48** | **88.70** | 91.30 |
| | Qwen3-14B | origin | 86.67 | 87.96 | 86.30 | 90.93 |
| | | CDS | 85.56 | 87.85 | **87.78** | 90.74 |
| | | CDS$_{bge}$ | 85.37 | 87.78 | **87.22** | **91.30** |
| geometry | gpt-5.2 | origin | 75.99 | 74.74 | 75.37 | 75.78 |
| | | CDS | **81.21** | 78.08 | **80.79** | **75.99** |
| | | CDS$_{bge}$ | **80.37** | 78.29 | 76.83 | 75.11 |
| | Qwen3-14B | origin | 66.18 | 65.76 | 65.14 | 73.07 |
| | | CDS | 65.55 | **68.27** | **68.89** | **73.90** |
| | | CDS$_{bge}$ | **66.60** | **67.85** | **70.36** | **74.32** |
| DetectiveQA | gpt-5.2 | origin | 80.52 | 82.47 | 83.77 | 85.71 |
| | | CDS | 80.52 | **83.12** | **85.06** | **88.31** |
| | | CDS$_{bge}$ | **81.17** | **85.71** | **86.36** | **88.31** |
| | Qwen3-14B | origin | 75.97 | 74.03 | 70.78 | 72.73 |
| | | CDS | **76.62** | 75.32 | **73.38** | **75.32** |
| | | CDS$_{bge}$ | 75.32 | **77.29** | **75.32** | **76.63** |

*Table 3.* CDS robustness across tasks, embedding models, and target LLMs. CDS uses the original embedding model, while CDS$_{bge}$ replaces it with bge-m3.

$(i, j)$, let

$$\delta_{ij} = \frac{\|\mathbf{e}_j - \mathbf{e}_i\|}{\max_{a,b} \|\mathbf{e}_b - \mathbf{e}_a\| + \epsilon}$$

be the normalized Euclidean distance, where $\epsilon$ is a small constant for numerical stability. Let

$$k(i, j) = \arg \min_{\ell \notin \{i,j\}} \left\| \mathbf{e}_\ell - \frac{\mathbf{e}_i + \mathbf{e}_j}{2} \right\|$$

be the demonstration nearest to the midpoint of $\mathbf{e}_i$ and $\mathbf{e}_j$, excluding the endpoints. We define the curvature proxy

$$\gamma_{ij} = \frac{1}{\pi} \arccos \left( \frac{(\mathbf{e}_j - \mathbf{e}_i)^\top (\mathbf{e}_{k(i,j)} - \mathbf{e}_j)}{\|\mathbf{e}_j - \mathbf{e}_i\| \, \|\mathbf{e}_{k(i,j)} - \mathbf{e}_j\|} \right),$$

with the cosine clipped to $[-1, 1]$ in implementation. The CDS edge cost is

$$D_{\mathrm{CDS}}(i, j) = \delta_{ij} + \gamma_{ij}.$$

The Euclidean component keeps adjacent demonstrations in related local neighborhoods, while the curvature proxy discourages sharp conceptual turns. We build a complete graph under this cost and compute a short tour using nearest-neighbor initialization followed by 2-opt local search (Croes, 1958). We then break the tour at its longest edge to obtain a path, and choose the best path across up to ten starting points according to the final smoothness score $1/(1 + \bar{\theta})$. Our theoretical claim only requires a sufficiently smooth pedagogical progression, not the global minimum of Eq. (4); empirically, this approximation is effective and remains inexpensive, taking under one minute on a standard CPU for $n \leq 128$.

| Task | Model | Method | Number of demonstrations | | | |
|---|---|---|---|---|---|---|
| | | | 16 | 32 | 64 | 128 |
| number_theory | gpt-5.2 | CDS | **88.15** | **91.48** | **88.70** | **91.30** |
| | | high curv | 85.74 | 89.63 | 86.48 | 88.15 |
| | Qwen3-14B | CDS | **85.37** | **87.78** | **87.22** | **91.30** |
| | | high curv | 79.26 | 84.44 | 84.26 | 90.37 |
| geometry | gpt-5.2 | CDS | **80.37** | **78.29** | **76.83** | **75.11** |
| | | high curv | 72.65 | 73.90 | 76.33 | 74.53 |
| | Qwen3-14B | CDS | **66.60** | **67.85** | **70.36** | **74.32** |
| | | high curv | 66.38 | 66.81 | 66.60 | 71.80 |

*Table 4.* Controlled smoothness ablation with bge-m3 embeddings. The same demonstrations are used for both orderings; only the transition curvature objective is inverted.

**High-curvature control.** To isolate curvature from local clustering, we compare CDS with a high-curvature ordering constructed from the same demonstrations. Both use Euclidean proximity, but the high-curvature variant inverts the curvature objective:

$$D_{\text{high curv}}(i, j) = \delta_{ij} + \left( \max_{a,b} \gamma_{ab} - \gamma_{ij} \right)$$

Thus, it still groups semantically related demonstrations while forcing abrupt turns between consecutive examples.

**Result.** We evaluate CDS on three reasoning tasks: geometry proof generation, number theory problem solving, and DetectiveQA logical reasoning. We further test robustness by replacing the original embedding model with bge-m3 and by evaluating an additional closed-source model, gpt-5.2. Table 3 shows that the gains persist across embedding models and target LLMs, especially on geometry and DetectiveQA; number_theory shows smaller margins because baseline accuracies are already high, leaving less room for ordering to improve performance. The performance gains also depend on the curvature of the original ordering. Our ablation against a high-curvature baseline strengthens our curvature claim.

## 7. Conclusion

Many-shot ICL has been largely understood through non-reasoning tasks, where scaling demonstrations is stable and ordering effects often fade. Our results show that these regularities do not transfer to many-shot CoT-ICL for reasoning: scaling depends on the model and task, similarity retrieval can fail due to procedural mismatch, and order variance grows with more CoT demonstrations. We therefore frame many-shot CoT-ICL as in-context test-time learning, where useful demonstrations should be understandable to the target model and arranged in a smooth knowledge progression. Guided by this view, CDS orders demonstrations by minimizing conceptual curvature and yields consistent gains across math and narrative reasoning, suggesting that reasoning trajectories can serve as reusable procedural guidance for future retrieval and prompting.

## Impact Statement

This paper presents work whose goal is to advance the field of Machine Learning. There are many potential societal consequences of our work, none which we feel must be specifically highlighted here.

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

# A. Details for Similarity-Based Demonstration Selection

This appendix provides implementation details for the similarity-based demonstration selection experiments in Section 4.3.

## A.1. Candidate pool and data splits

For each task, we form a demonstration candidate pool from the task's training split. All test queries are drawn from the task's test split. Since candidate demonstrations and evaluated queries come from disjoint splits, there is no overlap between a test query and any candidate demonstration.

## A.2. Embedding model and similarity

We embed each *question* (not the answer or rationale) using Qwen3-Embedding-4B (Zhang et al., 2025). Let $e(q) \in \mathbb{R}^d$ denote the embedding of a test question and $e(x) \in \mathbb{R}^d$ the embedding of a candidate training question. We measure semantic similarity by cosine similarity:

$$s(q,x) = \frac{e(q)^\top e(x)}{\|e(q)\|\|e(x)\|}.$$

For each test query $q$, we rank all candidates $x$ by $s(q,x)$.

## A.3. Constructing the most-similar and most-dissimilar sets

Given a target number of demonstrations $k$, we construct: (i) the **most-similar** set by selecting the top-$k$ candidates under $s(q,x)$, and (ii) the **most-dissimilar** set by selecting the bottom-$k$ candidates.

Unless otherwise stated, we present the selected examples to the LLM in descending order of similarity for the most-similar set, and ascending order of similarity for the most-dissimilar set.

## A.4. Qualitative Example: When "Similar" Questions Provide Misleading CoT

Table 5 shows an illustrative failure case from a reasoning task. Although the retrieved demonstration question is highly similar under embedding similarity, its solution uses a different invariant/decomposition than the query. When the LLM is conditioned on this demonstration, it tends to reuse the same intermediate steps, leading to an incorrect conclusion. In contrast, a less similar (but structurally closer) demonstration encourages the correct decomposition and improves accuracy.

**Test query (geometry).**
```
In the diagram, $\triangle XYZ$ is right-angled at $X,$ with
    $YX=60$ and $XZ=80.$ The point $W$ is on $YZ$ so that
    $WX$ is perpendicular to $YZ.$ Determine the length of
    $WZ.$ [asy]
pair X, Y, Z, W;
Y=(0,0);
X=(36,48);
Z=(100,0);
W=(36,0);
draw(X--Y--Z--X--W);
label("Y", Y, SW);
label("X", X, N);
label("W", W, S);
label("Z", Z, SE);
label("60", (X+Y)/2, NW);
label("80", (X+Z)/2, NE);
{[/asy]}\\
```

**Solution.**
```
By the Pythagorean Theorem, \begin{align*}
YZ^2 &= YX^2 + XZ^2
&= 60^2+80^2
&= 3600+6400
&=10000,
\end{align*} so $YZ=100.$

(We could also have found $YZ$ without using the Pythagorean
    Theorem by noticing that $\triangle XYZ$ is a right-
    angled triangle with its right-angle at $X$ and $XY=60=3\
    cdot 20$ and $XZ=80=4\cdot 20.$ This means that $\
    triangle XYZ$ is similar to a 3-4-5 triangle, and so $YZ
    =5\cdot 20=100.$)

Since $\triangle YXZ$ is right-angled at $X,$ its area is $$\
    frac{1}{2}\cdot 60\cdot 80=2400.$$ Since $XW$ is
    perpendicular to $YZ,$ then the area of $\triangle YXZ$
    is also equal to $$\frac{1}{2}\cdot 100\cdot XW=50XW.$$
    Therefore, $50XW=2400,$ so $XW=48.$ By the Pythagorean
    Theorem, \begin{align*}
WZ^2 &= 80^2 - 48^2
&= 6400 - 2304
&= 4096.
\end{align*} Thus, $WZ = \sqrt{4096}=\boxed{64}.$

An alternative solution comes by noticing that $\triangle XZW$
    and $\triangle YZX$ are similar. Therefore \[\frac{WZ}{XZ
    }=\frac{XZ}{YZ}\] or \[\frac{WZ}{80}=\frac{80}{100}=\
    frac45.\] This tells us that  \[WZ=\frac45\cdot80=\boxed
    {64}.\]
```

---

**Similar demonstration with cosine similarity.**
**Test query.**
```
In the diagram, $\triangle ABE$, $\triangle BCE$ and $\triangle
    CDE$ are right-angled, with $\angle AEB=\angle BEC = \
    angle CED = 60^\circ$, and $AE=24$. [asy]
pair A, B, C, D, E;
A=(0,20.785);
B=(0,0);
C=(9,-5.196);
D=(13.5,-2.598);
E=(12,0);
draw(A--B--C--D--E--A);
draw(B--E);
draw(C--E);
label("A", A, N);
label("B", B, W);
label("C", C, SW);
label("D", D, dir(0));
label("E", E, NE);
[/asy] Find the length of $CE$.
```

**Solution.**
```
We find $CE$ by first finding $BE$.

Since $AE = 24$ and $\angle AEB = 60^\circ$ and $AEB$ is a
    right triangle, then we can see that $AE$ is the
    hypotenuse and $BE$ is the shorter leg, so $BE = \dfrac
    {1}{2} \cdot 24 = 12.$

Likewise, since $BE = 12$ and $\angle BEC = 60^\circ$, then $CE
    = \dfrac{1}{2} \cdot 12 = \boxed{6}$.
```

**Why it is misleading.**
Since the retrieved example uses a $30^\circ-60^\circ-90^\circ$ triangle with the $1:2$ ratio and never uses an altitude-to-hypotenuse configuration, its method does not transfer.

*Table 5.* A qualitative example illustrating why question-level semantic similarity selects demonstrations with incompatible reasoning trajectories.

## A.5. Analysis of Similarity in Different LLM Types

Figures 9 and 10 compare similarity-based selection across non-reasoning and reasoning LLMs separately.

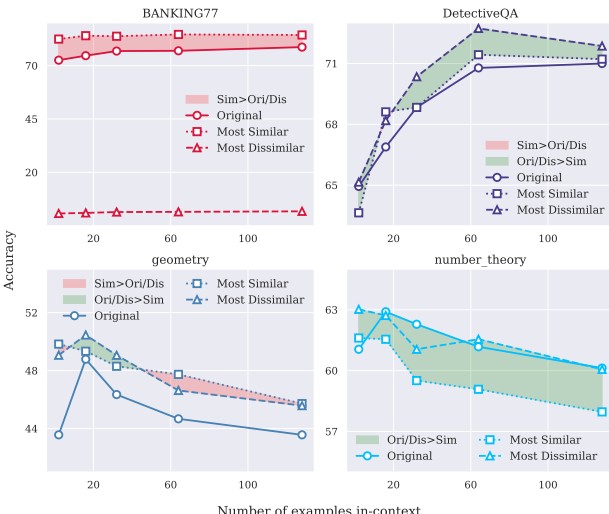

*Figure 9.* Performance with original (ori), similarity(sim) and dissimilar(dis) sets averaged across *three non-reasoning LLMs*. The area between the two sets is filled with colors, indicating the relative performance at each point.

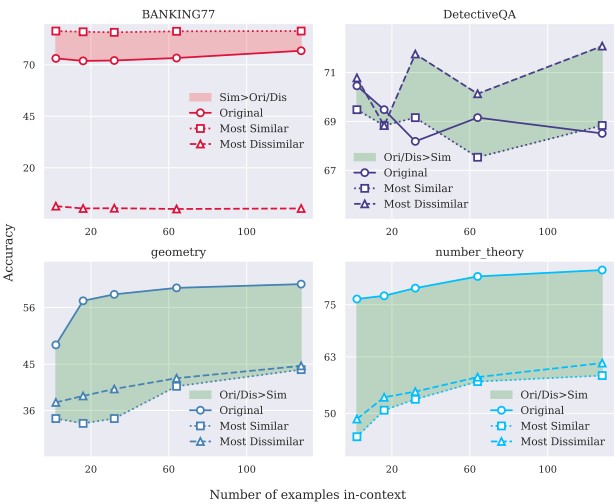

*Figure 10.* Performance with original (ori), similarity(sim) and dissimilar(dis) sets averaged across *two reasoning LLMs*. The area between the two sets is filled with colors, indicating the relative performance at each point.

## B. Statistical Robustness on a New ICL Subset

We compute the mean and standard deviation across five random demonstration-ordering seeds, repeating the analysis on a newly sampled ICL subset. These results strengthen the claims in Figures 7, 8, and 6: the observed trends persist beyond a single ordering or candidate pool.

| Model | Shots | Origin ($\mu \pm \sigma$) | First ($\mu \pm \sigma$) |
|---|---|---|---|
| Qwen3-14B | 16 | $86.33 \pm 0.86$ | $\mathbf{89.33} \pm 1.28$ |
| | 32 | $88.07 \pm 0.34$ | $\mathbf{92.19} \pm 0.50$ |
| | 64 | $88.85 \pm 0.80$ | $\mathbf{93.30} \pm 0.73$ |
| | 128 | $91.19 \pm 0.76$ | $\mathbf{94.44} \pm 0.57$ |
| Qwen3-8B | 16 | $83.48 \pm 0.81$ | $\mathbf{83.63} \pm 0.79$ |
| | 32 | $85.19 \pm 0.51$ | $\mathbf{87.41} \pm 0.74$ |
| | 64 | $88.04 \pm 0.84$ | $\mathbf{91.08} \pm 0.69$ |
| | 128 | $86.93 \pm 1.42$ | $\mathbf{88.52} \pm 0.49$ |

*Table 6.* Reasoning-oriented LLMs on number_theory across five random ordering seeds.

| Model | Shots | Origin ($\mu \pm \sigma$) | Wrong ($\mu \pm \sigma$) |
|---|---|---|---|
| Llama-3.1-8B | 16 | $23.55 \pm 3.58$ | $\mathbf{35.57} \pm 0.72$ |
| | 32 | $22.21 \pm 8.08$ | $\mathbf{35.91} \pm 1.13$ |
| | 64 | $25.97 \pm 6.42$ | $\mathbf{35.61} \pm 1.62$ |
| | 90 | $32.78 \pm 6.29$ | $\mathbf{36.12} \pm 1.16$ |
| Qwen2.5-14B | 16 | $58.37 \pm 0.83$ | $\mathbf{61.05} \pm 1.19$ |
| | 32 | $59.00 \pm 0.91$ | $\mathbf{59.67} \pm 0.97$ |
| | 64 | $57.70 \pm 1.13$ | $\mathbf{60.75} \pm 1.25$ |
| | 90 | $57.41 \pm 1.58$ | $\mathbf{59.46} \pm 0.70$ |

*Table 7.* Non-reasoning LLMs on geometry across five random ordering seeds.

| Task | Target model | Shots | Stronger LLM | Self-generated |
|---|---|---|---|---|
| number_theory | Qwen3-8B | 16 | $\mathbf{86.26} \pm 0.61$ | $83.63 \pm 0.79$ |
| | | 32 | $\mathbf{89.55} \pm 1.00$ | $87.41 \pm 0.74$ |
| | | 64 | $88.19 \pm 1.63$ | $\mathbf{91.08} \pm 0.69$ |
| | | 128 | $81.15 \pm 2.01$ | $\mathbf{88.52} \pm 0.49$ |
| geometry | Llama-3.1-8B | 16 | $33.53 \pm 2.05$ | $\mathbf{35.57} \pm 0.72$ |
| | | 32 | $34.28 \pm 2.06$ | $\mathbf{35.91} \pm 1.13$ |
| | | 64 | $35.61 \pm 1.05$ | $35.61 \pm 1.62$ |
| | | 90 | $33.48 \pm 0.52$ | $\mathbf{36.12} \pm 1.16$ |

*Table 8.* CoT-ICL generated from stronger LLMs versus self-generated demonstrations across five random ordering seeds. Values report $\mu \pm \sigma$.

## C. Curvature-based Smoothness: Details and Implementation

To quantify the relationship between demonstration ordering smoothness and ICL performance, we develop Algorithm 1. The algorithm takes as input multiple orderings of demonstrations and their corresponding performance scores, and outputs a correlation coefficient between ordering smoothness and performance.

### C.1. Demonstration Format

For the smoothness analysis in Section 5.2, each demonstration is embedded as a single text string that concatenates `Question + Chain-of-Thought + Answer`. We use the same template in Appendix E for constructing demonstrations. All curvature results in the main paper use this template unless stated otherwise.

**Algorithm 1** Curvature–Performance Correlation Analysis

**Input:** $k$ orderings $\{E^{(j)}\}_{j=1}^{k}$, where $E^{(j)} = [\mathbf{e}_1^{(j)}, \ldots, \mathbf{e}_N^{(j)}]^{\top}$; performance scores $S = [S_1, \ldots, S_k]$
**Output:** Pearson correlation coefficient $r$ between smoothness scores $\mathbf{m}$ and performance $S$
Initialize smoothness scores $\mathbf{m} \leftarrow [0, \ldots, 0]$ (length $k$)
**for** $j = 1$ **to** $k$ **do**
   Initialize curvature list $\Theta \leftarrow [\,]$
   **for** $i = 2$ **to** $N - 1$ **do**
      $\mathbf{v}_1 \leftarrow \mathbf{e}_i^{(j)} - \mathbf{e}_{i-1}^{(j)}$
      $\mathbf{v}_2 \leftarrow \mathbf{e}_{i+1}^{(j)} - \mathbf{e}_i^{(j)}$
      **if** $\|\mathbf{v}_1\| > 0$ **and** $\|\mathbf{v}_2\| > 0$ **then**
         $c \leftarrow \dfrac{\mathbf{v}_1^{\top} \mathbf{v}_2}{\|\mathbf{v}_1\| \, \|\mathbf{v}_2\|}$
         $c \leftarrow \min(1, \max(-1, c))$
         $\theta \leftarrow \arccos(c)$
         Append $\theta$ to $\Theta$
      **end if**
   **end for**
   $\bar{\theta}^{(j)} \leftarrow \mathrm{mean}(\Theta)$
   $\mathbf{m}[j] \leftarrow \dfrac{1}{1 + \bar{\theta}^{(j)}}$
**end for**
$r \leftarrow \mathrm{PearsonCorrelation}(\mathbf{m}, S)$
**return** $r$

### C.2. Embedding Model

Similar to section 4.3, we encode demonstrations using Qwen3-Embedding-4B (Zhang et al., 2025). Let $\mathbf{d}_i$ denote the formatted demonstration string and $\mathrm{Embed}(\cdot)$ the embedding model. We obtain vectors $\mathbf{e}_i = \mathrm{Embed}(\mathbf{d}_i) \in \mathbb{R}^d$ using the embedding model's default output representation with vLLM deployment (Kwon et al., 2023).

### C.3. Correlation Protocol

To study the relationship between ordering smoothness and downstream accuracy, we use the multiple random permutations of a fixed demonstration set with $n = 128$ demonstrations from Section 4.4. For each ordering $O$, we compute the bounded smoothness score $m(O) = 1/(1 + \bar{\theta}(O))$ and evaluate task accuracy under the corresponding prompt. We then compute the Pearson correlation between $\{m(O)\}$ and accuracy over the sampled orderings.

## D. CDS: Details and Implementation

### D.1. Model Studies

We design our experiments to isolate the effect of *demonstration ordering* in many-shot CoT-ICL. To this end, we focus on reasoning-oriented LLMs that exhibit a positive scaling trend with more demonstrations, since such models demonstrate in-context learning capacity and should benefit from improved ordering.

```
Given a query, answer yes or no to the query.
The predicted answer must come from the demonstration examples
    with the exact format. The examples are as follows:
Question: In the sentence "{text_1}", does the pronoun "{
    span2_text_1}" refer to {span1_text_1}?
Answer: {answer_1}
...
Question: In the sentence "{text_n}", does the pronoun "{
    span2_text_n}" refer to {span1_text_n}?
Answer: {answer_n}

Now predict the answer for the following query:
Question: In the sentence "{text_i}", does the pronoun "{
    span2_text_i}" refer to {span1_text_i}?
Reply in the following format:
Answer: [yes | no]
```

*Figure 11.* Prompt for WSC task

We also control for confounding factors unrelated to ordering quality by using dataset-provided CoT rationales and answers, avoiding performance degradation due to incorrect or low-quality generated rationales (i.e., self-generated CoT in section 5.1). Under this setup, we study models that can interpret and leverage the provided CoT.

Our primary evaluation uses Qwen3 (8B and 14B) across varying numbers of demonstrations and three tasks (geometry, number theory, and DetectiveQA), as these models satisfy the above criteria and provide a stable platform for many-shot experiments.

## E. Prompt Formatting for Each Task

### E.1. SuperGlue

We evaluate the Winograd Schema Challenge (WSC) for coreference resolution, and the Choice of Plausible Alternatives (COPA) for open-domain commonsense causal reasoning. Both are formatted as a binary-label classification task. The prompt for inference is presented in Figure 11.

```
Answer in A or B.
The predicted answer must come from the demonstration examples
    with the exact format. The examples are as follows:
Premise: {premise_1}
Question: What is the {question_1} for this?
Options:
A. {choice1_1}
B. {choice2_1}
Answer: {answer_1}
...
Premise: {premise_n}
Question: What is the {question_n} for this?
Options:
A. {choice1_n}
B. {choice2_n}
Answer: {answer_n}

Now predict the answer for the following query:
Premise: {premise_i}
Question: What is the {question_i} for this?
Options:
A. {choice1_i}
B. {choice2_i}
Reply in the following format:
Answer: [A | B]
```

*Figure 12.* Prompt for COPA task

## E.2. TREC

We evaluate the Text REtrieval Conference (TREC) Question Classification dataset with 50 fine class labels. The prompt for inference is presented in Figure 13.

```
Given a question, predict the label of the question. You can only
    make predictions from the following categories: {
    LIST_OF_CATEGORIES}
Please predict the label of the FINAL question with the provided
    demonstration example queries as follows:
question: {question_1}
label: {label_1}
...
question: {question_n}
label: {label_n}

Now predict the answer for the following query:
question: {question_i}

Reply in the following format:
label: [category_name]
```

*Figure 13.* Prompt for TREC task

## E.3. BANKING77

We evaluate the BANKING77 dataset with 77 fine-grained intents in the banking domain. The prompt for inference is presented in Figure 14.

```
Given a question, predict the label of the question. You can only
    make predictions from the following categories: {
    LIST_OF_CATEGORIES}
Please predict the intent category of the FINAL query with the
    provided demonstration example queries as follows:
service query: {question_1}
intent category: {label_1}
...
service query: {question_n}
intent category: {label_n}

Now predict the intent category for the following query:
service query: {question_i}

Reply in the following format:
intent category: [category_name]
```

*Figure 14.* Prompt for BANKING77 task

## E.4. NLU

We evaluate the NLU dataset with 68 fine-grained intents in the conversational domain. The prompt for inference is presented in Figure 15.

## E.5. GSM8K

We evaluate the GSM8K dataset for grade school math word problems. The prompt for inference is presented in Figure 16.

## E.6. MATH

We evaluate the Mathematics Aptitude Test of Heuristics (MATH) dataset for mathematics competition problems, including the question types of counting_and_probability, pre-algebra, geometry, precalculus, number_theory and algebra.

```
Given a question, predict the label of the question. You can only
    make predictions from the following categories: {
    LIST_OF_CATEGORIES}
Please predict the intent category of the FINAL utterance with
    the provided demonstration example queries as follows:
utterance: {question_1}
intent category: {label_1}
...
utterance: {question_n}
intent category: {label_n}

Now predict the intent category for the following utterance:
utterance: {question_i}

Reply in the following format:
intent category: [category_name]
```

*Figure 15.* Prompt for NLU task

```
In the end of the response, add a summary 'The answer is [answer
    ].'
Q: {question_1}
A: {CoT_1} {answer_1}
...
Q: {question_n}
A: {CoT_n} {answer_n}

### Q: {question_t}
### A: Let's think step by step.
```

*Figure 16.* Prompt for GSM8K task

The prompt for inference is presented in Figure 17.

```
Write a response that appropriately completes the request and
    wrap the final answer inside \boxed{}.
Problem: {question_1}
Solution: {CoT_with_answer_1}
...
Problem: {question_n}
Solution: {CoT_with_answer_n}

### Problem: {question_t}
### Solution: Let's think step by step.
```

*Figure 17.* Unified prompt for MATH task

## E.7. DetectiveQA

DetectiveQA (Xu et al., 2024) is a long-context narrative reasoning benchmark. Each instance includes an *evidence* section as part of the input, and the goal is to answer a question by reasoning over this evidence. DetectiveQA additionally provides annotated reasoning chains for deriving the answer from the evidence. When constructing CoT demonstrations, we use the derivation labeled "-1" as the corresponding chain-of-thought.

To avoid potential information leakage, we further filter the test split. We exclude any test instances that share the same data source (i.e., novel ID) with the training split. This prevents the model from receiving extra clues through CoT-ICL demonstrations drawn from the same underlying narrative. The prompt for inference is presented in Figure 18.

```
Below is an instruction that describes a task.\n Select the
      correct option from A/B/C/D. Answer with 'The answer is {A/
      B/C/D}.' in the end of your response.\n\n"

Question: {question_1}
Context: {context_1}
Options:
A. {option_1_1}
B. {option_1_2}
C. {option_1_3}
D. {option_1_4}
Answer:
{derivation_1}
The answer is {answer_1}.

...

Question: {question_n}
Context: {context_n}
Options:
A. {option_n_1}
B. {option_n_2}
C. {option_n_3}
D. {option_n_4}
Answer:
{derivation_n}
The answer is {answer_n}.

### Question: {question_i}
### Context: {context_i}
### Options:
A. {option_i_1}
B. {option_i_2}
C. {option_i_3}
D. {option_i_4}
### Answer:
```

*Figure 18.* Prompt for DetectiveQA task

