# OpenReview forum: "Many-Shot CoT-ICL: Making In-Context Learning Truly Learn"
_ICML.cc/2026/Conference — ICML 2026 regular_

### Official Review · Reviewer_uvmk · 2026-03-08

**Soundness:** 2
**Presentation:** 3
**Significance:** 2
**Originality:** 3
**Overall Recommendation:** 4
**Confidence:** 3

**Summary:**

This paper investigates the scaling behavior of many-shot Chain-of-Thought (CoT) in-context learning for reasoning tasks. The authors empirically observe that simply adding more CoT demonstrations can degrade performance for non-reasoning models, and that standard semantic similarity retrieval fails to generalize to complex reasoning. To address this, they propose Curvilinear Demonstration Selection (CDS), a sequence ordering method that treats prompt arrangement as a Traveling Salesperson Problem (TSP) in a PCA/UMAP-projected embedding space, claiming an average performance improvement of 3.81%.

**Compliance With Llm Reviewing Policy:**

Affirmed.

**Final Justification:**

This paper presents a interesting approach to ordering many-shot CoT demonstrations by minimizing latent space curvature. While my initial review highlighted weaknesses regarding statistical soundness (lack of error bars) and practical significance (the high cost of deploying 100+ CoT examples), the authors' rebuttal effectively addressed these core issues. The newly provided multi-seed variance analyses and robustness checks using alternative embeddings significantly strengthen the paper's overall soundness. Additionally, framing the massive context injection as analogous to tool/skill packing in modern Agent architectures successfully clarifies its real-world significance. Overall, the authors' diligent rebuttal has resolved my primary concerns and reinforced the work's originality and practical value, shifting my final evaluation to a Weak Accept (4).

**Key Questions For Authors:**

1. Statistical Significance: Could you kindly provide error bars or standard deviations across multiple evaluation runs (e.g., using different random seeds or prompt sub-samples) for the experiments in Figures 7 and 8? This would help confirm that the fascinating preference for "wrong" CoTs is a robust phenomenon rather than an artifact of prompt variance.

2. Validity of the Metric: Could you elaborate on how spatial curvature in a standard textual embedding model effectively captures "procedural compatibility" or algorithmic reasoning steps, as opposed to just superficial topic or keyword clustering?

3. Real-world Use Case & Cost-Benefit: Fully acknowledging the benefits of KV cache reuse, in what specific practical scenarios do you envision this many-shot CoT paradigm being more advantageous than improving the base model's fundamental reasoning capabilities via RL or fine-tuning with high-quality data?

**Limitations:**

No. The authors did not adequately discuss the vulnerability of their approach to high variance, nor did they critically address the disconnect between semantic embeddings and logical reasoning trajectories.

**Strengths And Weaknesses:**

**Strengths**:

1. Counter-intuitive & Valuable Insight: The finding that self-generated, albeit incorrect, reasoning trajectories can sometimes outperform gold-standard rationales is highly novel and thought-provoking. It highlights a critical insight for the ICL community: distributional alignment and stylistic familiarity can occasionally outweigh absolute logical correctness when conditioning models.

2. Interesting Empirical Observations: The paper provides a useful empirical data point by highlighting that standard semantic similarity retrieval (which is common in RAG setups) tends to fail specifically for complex CoT reasoning tasks.

3. Creative Formulation: Framing the context ordering problem as a spatial path-smoothing problem (TSP) in a latent space is a mathematically elegant and creative approach to prompt engineering.

**Weaknesses**:

1. Abstraction of the Smoothness Metric: While the CDS method is mathematically elegant, the core assumption—that minimizing spatial curvature in a PCA/UMAP-reduced semantic embedding space equates to "smooth logical or procedural progression"—might be somewhat of a leap. It would be beneficial to see more discussion on whether standard textual embeddings truly capture underlying mathematical algorithmic steps rather than just surface-level semantic clustering.

2. Need for Statistical Rigor on Key Claims: The overall performance improvement (averaging 3.81%) is relatively modest. More importantly, for the fascinating claim that models perform better with "wrong" self-generated CoTs (Figures 7 & 8), the current draft seems to lack error bars or variance analysis. Given that the paper itself nicely demonstrates the high sensitivity of many-shot CoT to prompt ordering, providing variance metrics would greatly strengthen the reliability of these specific findings.

3. Exploration of Practical Deployment Scenarios: I acknowledge the technical advantage of many-shot ICL—specifically that the KV cache for a massive static context of 100+ CoT demonstrations can be computed once and reused across multiple test queries. However, I am genuinely curious about its optimal real-world deployment scenarios. With the community currently finding substantial success in enhancing fundamental reasoning via Reinforcement Learning (RL) and scaling high-quality training data, the long-term cost-benefit ratio of relying on extremely long in-context examples at inference time might benefit from further clarification.

---

> ### Author Rebuttal · Authors · 2026-03-31
>
> We sincerely thank the reviewer for raising insightful reviews, we will address your concern in detail,
>
> ### Abstraction of the Smoothness Metric
> To effectively capture procedural compatibility and algorithmic steps, our smoothness metric and CDS method explicitly embed the full demonstration (Question + Chain-of-Thought + Answer). By incorporating the CoT rationale into the embedding process, the representations capture the specific logical structures and mathematical operations utilized in the solutions. While retrieving via full-demonstration similarity is ideal, the ground-truth CoT is unknown during testing. Alternatively, retrieving based on the model's self-generated CoT also helps: for \(n=1\) (avoiding ordering effects), accuracy on number_theory and geometry improves from 78.52% to 80.19% and from 56.58% to 56.99% with Qwen3-8B, while doubling the inference cost.
>
> ### Need for Statistical Rigor on Key Claims
> We agree that providing variance metrics would greatly strengthen the reliability. To address this, we have computed the mean and standard deviation across 5 different random demonstration ordering seeds. To further demonstrate the robustness of our findings, we experiment with a different pool of training samples to construct CoT-ICL to see whether this observation persists.
>
> **1. Reasoning-Oriented LLMs**
>
> **task: number_theory**
> | Model | Shots | Origin ($\mu \pm \sigma$) | First ($\mu \pm \sigma$) |
> | :--- | :---: | :--- | :--- |
> | **Qwen3-14B** | 16 | $86.33 \pm 0.86$ | $89.33 \pm 1.28$ |
> | | 32 | $88.07 \pm 0.34$ | $92.19 \pm 0.50$ |
> | | 64 | $88.85 \pm 0.80$ | $93.30 \pm 0.73$ |
> | | 128 | $91.19 \pm 0.76$ | $94.44 \pm 0.57$ |
> | **Qwen3-8B** | 16 | $83.48 \pm 0.81$ | $83.63 \pm 0.79$ |
> | | 32 | $85.19 \pm 0.51$ | $87.41 \pm 0.74$ |
> | | 64 | $88.04 \pm 0.84$ | $91.08 \pm 0.69$ |
> | | 128 | $86.93 \pm 1.42$ | $88.52 \pm 0.49$ |
>
> **2. Non-Reasoning LLMs**
>
> **task: geometry**
> | Model | Shots | Origin ($\mu \pm \sigma$) | Wrong ($\mu \pm \sigma$) |
> | :--- | :---: | :--- | :--- |
> | **Llama-3.1-8B** | 16 | $23.55 \pm 3.58$ | $35.57 \pm 0.72$ |
> | | 32 | $22.21 \pm 8.08$ | $35.91 \pm 1.13$ |
> | | 64 | $25.97 \pm 6.42$ | $35.61 \pm 1.62$ |
> | | 90 | $32.78 \pm 6.29$ | $36.12 \pm 1.16$ |
> | **Qwen2.5-14B** | 16 | $58.37 \pm 0.83$ | $61.05 \pm 1.19$ |
> | | 32 | $59.00 \pm 0.91$ | $59.67 \pm 0.97$ |
> | | 64 | $57.70 \pm 1.13$ | $60.75 \pm 1.25$ |
> | | 90 | $57.41 \pm 1.58$ | $59.46 \pm 0.70$ |
>
>
> **3. CoT-ICL generated from Stronger LLMs vs. Self-Generated**
>
> | Task & Target Model | Shots | From Stronger LLM's ($\mu \pm \sigma$) | Self-Generated ($\mu \pm \sigma$) |
> | :--- | :---: | :--- | :--- |
> | **number theory** | 16 | $86.26 \pm 0.61$  | $83.63 \pm 0.79$  |
> | *(Qwen3-8B)*| 32 | $89.55 \pm 1.00$  | $87.41 \pm 0.74$  |
> | | 64 | $88.19 \pm 1.63$  | $91.08 \pm 0.69$  |
> | | 128| $81.15 \pm 2.01$  | $88.52 \pm 0.49$  |
> | **geometry** | 16 | $33.53 \pm 2.05$  | $35.57 \pm 0.72$  |
> | *(Llama-3.1-8B)*| 32 | $34.28 \pm 2.06$  | $35.91 \pm 1.13$  |
> | | 64 | $35.61 \pm 1.05$  | $35.61 \pm 1.62$  |
> | | 90 | $33.48 \pm 0.52$  | $36.12 \pm 1.16$  |
>
> Experiment results strengthen our claims in Figures 7 and 8.
>
> ### Concern for Real-world Use Case & Cost-Benefit
> To address this concern, our work can benefit from the below points,
> - We envision the Many-Shot CoT paradigm being particularly advantageous over RL or fine-tuning in scenarios that require rapid and low-cost customization. For example, in chat UI, different users and varied windows demand highly customized use cases. Injecting ICL serves as a low-cost, feasible, and immediate solution.
> - Our findings (e.g., the gains of self-generated demonstrations and the effectiveness of CDS), hold true even at much smaller scales (e.g., n=16).
> - By better absorbing task procedures, models can leverage procedural knowledge for future retrieval (similar to ExpeL using previous trajectories). The "thinking" content can similarly act as crucial procedural guidance. To support this, we further compared performance at $n=128$ (i.e., the context length that achieves near-optimal performance) between enabling and disabling the thinking process.
>
>     | Model | geometry | number_theory | DetectiveQA |
>     |-------|----------|---------------|-------------|
>     | Qwen3-14B (enable) | 73.07 | 91.30 | 72.73 |
>     | Qwen3-14B (disable) | 65.76 | 88.15 | 72.73 |
>     | Qwen3-8B (enable) | 67.01 | 84.63 | 69.48 |
>     | Qwen3-8B (disable) | 62.63 | 79.81 | 66.88 |
>
>     With thinking disabled, models typically fail catastrophically. The context within the thinking token provides the procedural understanding to LLM. To further investigate, the experiment shows that as $n$ increases from 16 to 128 for Qwen3-8B on geometry, accuracy rises from 66.18% to 73.07%, while the average length of the generated \<think\> tokens decreases by 24.02%. This demonstrates that the model internalizes the procedural schema from many-shot CoT.

---

> > ### Author Rebuttal · Reviewer_uvmk · 2026-04-01
> >
> > Thanks for the additional experiments. The added error bars and the robustness checks with bge-m3 address my previous concerns regarding variance and potential cherry-picking.
> >
> > I've also reconsidered the practical side: injecting 100+ CoT examples is actually quite similar to modern Agent setups where we often pack 100+ skills or tool descriptions into a long context. In that specific context, a stable ordering strategy like CDS has clear practical value. I'm raising my score to 4.

---

> > > ### Author Response · Authors · 2026-04-01
> > >
> > > Thank you so much for your review, your recognition in the practical side, and for raising your score! We will certainly include all these additional experiments in the revised version of the paper. Thanks again for your constructive and encouraging feedback!

---

### Official Review · Reviewer_zKhU · 2026-03-12

**Soundness:** 3
**Presentation:** 3
**Significance:** 3
**Originality:** 3
**Overall Recommendation:** 4
**Confidence:** 3

**Summary:**

This paper demonstrates that standard many-shot in-context learning scaling laws fail for reasoning tasks, where increasing the number of Chain-of-Thought (CoT) demonstrations leads to performance instability and high sensitivity to example ordering. To address this, the authors reframe many-shot CoT-ICL as "in-context test-time learning" rather than simple pattern matching, arguing that effective prompts must act as a structured curriculum that is both understandable to the model and conceptually smooth. The author introduces Curvilinear Demonstration Selection (CDS), a method that optimizes example sequencing by minimizing total curvature in embedding space, resulting in an average performance gain of 3.81% across various reasoning benchmarks.

**Compliance With Llm Reviewing Policy:**

Affirmed.

**Key Questions For Authors:**

See weaknesses above.

**Limitations:**

yes

**Strengths And Weaknesses:**

### Strengths

- The observations on experiments across both non-reasoning tasks and reasoning tasks support the claims of such scaling disparity.
- The author investigates why traditional methods fail, such as showing that semantic similarity is a weak proxy for CoT compatibility.
- The CDS method provides a concrete, principle-driven strategy for prompt engineering that practitioners can use to boost the reasoning capabilities of long-context models.

### Weaknesses

- The claim of in-context test-time learning is highly based on the observations
- The evaluation focuses exclusively on standard reasoning benchmarks and lacks a sensitivity analysis regarding the quality and logical coherence of the initial demonstration pool, which leaves the efficacy of the CDS reordering mechanism unexplored for scenarios involving procedurally disjoint examples or extreme distributional imbalances.
- The paper lacks sufficient technical detail and systematic ablation explorations for the CDS algorithm, as these critical components are disproportionately deferred to the supplementary material rather than being integrated into the main discussion.
- Minor typo in line 199, x-axis represents the number of in-context demonstrations, while y-axis is normalized accuracy.

---

> ### Author Rebuttal · Authors · 2026-03-31
>
> We sincerely thank the reviewer comment on how we can better structure our paper, since there is a page limitation in the current stage, leaving details in the appendix. We will include the core details of the algorithm in the main part in the revision. And we would  like to address your concern as follows,
>
> ### Claims are highly based on the observations
> We agree with the reviewer that our framing of many-shot CoT-ICL as "in-context test-time learning" is driven by our empirical observations. While we agree with this, the findings are not isolated anomalies, they form a cohesive set of evidence that contradicts the traditional "pattern matching" hypothesis and points directly to an active, test-time learning process. Our framing is built on four interconnected empirical observations:
>
> - The Failure of Semantic Similarity: If CoT-ICL were merely large-scale pattern matching, retrieving semantically similar examples should help. Instead, we observed the opposite for reasoning tasks, indicating the model is trying to extract a procedural algorithm rather than simply copying surface-level patterns.
> - Growing Path Dependence (Order Sensitivity): The growing instability for reasoning tasks along growing CoT-ICL suggests that the model is undergoing a path-dependent learning process.
> - The Necessity of "Understanding" (Distributional Alignment): We found that models perform better when prompted with their own self-generated rationales compared to dataset-provided gold rationales. This implies that demonstrations act as pedagogical supervision that the model must internalize, rather than just static text to copy.
> - The Need for Smooth Information Flow: We quantified the "smoothness" of the prompt by measuring the trajectory of demonstration embeddings. Just like a student learning from a well-structured curriculum, the LLM requires a smooth conceptual progression to build a reliable internal procedure.
>
> In summary, while our "test-time learning" perspective is empirically grounded, it is the only unified framework that adequately explains why similarity fails, why order matters more at scale, and why self-generated smooth trajectories perform best. We would be glad to add a discussion paragraph in the revised manuscript to bridge how these empirical observations necessitate our proposed conceptual framework.
>
> ### Efficacy study is lacking for CDS
> In this work, our primary objective in proposing CDS is to isolate and investigate the specific impact of smooth knowledge progression via demonstration ordering. If we were to deliberately include procedurally disjoint examples or extreme distributional imbalances into the initial pool, it would introduce confounding variables (acting as additional abrupt signals). Under such conditions, it would be difficult to conduct a controlled experiment to determine whether performance variations stem from the ordering mechanism itself or from the inherent noise of the demonstration pool.
>
> To evaluate CDS, we adopted the ground-truth CoTs provided in the datasets and maintained the exact same demonstration sets for both the origin baseline and CDS. This strict control ensures that the only variation between the two settings is the sequence order. Through this controlled setup, we are able to robustly establish the empirical finding that order does matter and is worth exploring for tasks that need complex reasoning or derivation.
>
> To solidify our claims, we additionally experimented with the following settings,
> - We replaced the original embedding model with `bge-m3` to extract features and perform the CDS (denoted as bge-CDS), comparing it against the original setup (CDS).
> - To ensure the performance gains of CDS are not limited to the Qwen family, we evaluated an additional closed-source model `gpt-5.2`.
>
> **geometry**
> | Model | Method | n=16 | n=16 | n=16 | n=16 |
> | :--- | :--- | :--- | :--- | :--- | :--- |
> | gpt-5.2 | origin | 75.99 | 74.74 | 75.37 | 75.78 |
> | gpt-5.2 | CDS | 81.21 | 78.08 | 80.79 | 75.99 |
> | gpt-5.2 | bge-CDS | 80.37 | 78.29 | 76.83 | 75.11 |
> | Qwen3-14B | origin | 66.18 | 65.76 | 65.14 | 72.03 |
> | Qwen3-14B | CDS | 65.55 | 68.27 | 68.89 | 73.90 |
> | Qwen3-14B | bge-CDS | 66.60 | 67.85 | 70.36 | 74.32 |
>
>
> **DetectiveQA**
> | Model | Method | n=16 | n=16 | n=16 | n=16 |
> | :--- | :--- | :--- | :--- | :--- | :--- |
> | gpt-5.2 | origin | 80.52 | 82.47 | 83.77 | 85.71 |
> | gpt-5.2 | CDS | 80.52 | 83.12 | 85.06 | 88.31 |
> | gpt-5.2 | bge-CDS | 81.17 | 85.71 | 86.36 | 88.31 |
> | Qwen3-14B | origin | 75.97 | 74.03 | 70.78 | 72.73 |
> | Qwen3-14B | CDS | 76.62 | 75.32 | 73.38 | 75.32 |
> | Qwen3-14B | bge-CDS | 75.32 | 77.29 | 75.32 | 76.63 |
>
> Our evaluations demonstrate consistent performance growth across different embedding models and different target evaluation LLMs, confirming that the smoothness principle is a robust, representation-independent property of in-context learning.

---

> > ### Author Rebuttal · Reviewer_zKhU · 2026-04-01
> >
> > I thank the authors for their rebuttal to address my concerns. The additional discussion and experimental analysis strengthen the robustness of this work. Please do include them in the revised manuscript. I would like to keep my positive score. Good luck.

---

> > > ### Author Response · Authors · 2026-04-02
> > >
> > > Thank you so much for your review and your positive score for our work! We will surely include all these additional discussions and experimental analyses in our revised manuscript!

---

### Official Review · Reviewer_X8ms · 2026-03-13

**Soundness:** 2
**Presentation:** 3
**Significance:** 2
**Originality:** 3
**Overall Recommendation:** 3
**Confidence:** 3

**Summary:**

This paper studies many-shot chain-of-thought in-context learning (CoT-ICL) and shows that the conclusions for standard ICL do not hold for CoT-ICL. For example, they find increasing the number of examples only helps for reasoning-oriented LLMs. They attribute these discrepancies to the fact that CoT-ICL is not pattern matching, but rather test-time learning. Building on this, they proposed a data selection and ordering method to improve CoT-ICL.

**Compliance With Llm Reviewing Policy:**

Affirmed.

**Final Justification:**

Many of the explanations in the paper remain largely qualitative and somewhat speculative. Moreover, evaluation is somewhat limited in scope, both in terms of baseline comparisons and diversity of tasks. I am keeping my score of 3.

**Key Questions For Authors:**

1. (minor) The x-axis and y-axis descriptions in Figure 2 are flipped.

2. In Figure 3, could you provide an intuition on why we see a drop in ICL accuracy for a non-reasoning LLM as we increase the number of samples? Moreover, why is this decreasing trend not observed in Figure 2 (right)?

3. Continuing the earlier question, can a non-reasoning LLM simply ignore the reasoning chain, essentially reducing the problem to a traditional ICL?

4. (minor) In the beginning of Section 5, it should be "Sections 4.3 and 4.4" instead of "Sections 4.4 and 4.3"

5. In Figure 7, the accuracy trend vs. number of demonstrations differ for each dataset. Is there any intuition behind this? Also, why do the accuracies tend to drop as the number of demonstrations increases? Lastly, why is the **wr** accuracy sometimes higher than the **cr** accuracy?

6. In Figure 8 (right), not only does the gain of **first** over **origin** diminish for the 14B model, it seems that **origin** is even better than **first**, especially for the DetectiveQA task. Could you provide any intuition?

**Limitations:**

Yes

**Strengths And Weaknesses:**

**Soundness:** This paper is fairly sound. Numerous experiments are conducted, but some of the results can be hard to interpret. For example, in Table 2, the supposed increase in accuracy on non-reasoning tasks is not apparent (e.g., performance often degrades with very high numbers of examples). Also, in Figure 4 (left), there does not seem to be an increasing trend in accuracy as a function of number of ICL demonstrations. See below for details.

**Presentation:** The paper is clearly written and easy to follow.

**Significance:**  This paper addresses a relevant problem on CoT-ICL, even though the potential use case is very limited.

**Originality:** This paper advances the field by rigorously studying the behavior of CoT-ICL and reframing it as in-context test-time learning to explain its discrepancy with standard ICL.

---

> ### Author Rebuttal · Authors · 2026-03-31
>
> We thank the reviewer for raising insightful reviews and pointing out the typo in our work, we will correct it accordingly. And we would like to address your concern in detail as below,
>
> ### Concern for significance
> Regarding the concern about the significance of our contributions and the potentially limited use cases, we would like to clarify our impact across two main dimensions:
> 1. Significance: Shifting from Pattern Matching to Genuine In-Context Learning
> We aim to present a general topic regarding how large language models reason and how In-Context Learning (ICL) transitions from shallow pattern matching to genuine in-context learning. To substantiate this, we present several interconnected empirical findings:
>     - Semantic Similarity Fails: Retrieving similar examples actually degrades reasoning performance, indicating models extract procedural algorithms rather than copying surface patterns.
>     - Growing Path Dependence: The growing instability suggests the model undergoes a highly path-dependent learning process.
>     - Necessity of "Understanding": Models perform best using self-generated rationales over gold rationales,  proving demonstrations act as internalizable supervision.
>     - Smooth Information Flow: LLMs require a smooth conceptual progression to build reliable internal procedures.
>
>
> 2. Use Cases
>
> Our findings extend far beyond many-shot scenarios:
> - Our findings (e.g., the gains of self-generated demonstrations and the effectiveness of CDS), hold true even at much smaller scales (e.g., n=16).
> - By better absorbing task procedures, models can leverage procedural knowledge for future retrieval (similar to ExpeL using previous trajectories). The "thinking" content can similarly act as crucial procedural guidance. To support this, we further compared performance at $n=128$ between enabling and disabling the thinking process. The results below show that enabling thinking consistently outperforms disabling it, highlighting its potential in constructing and transferring procedural guidance.
>
>     | Model | geometry | number_theory | DetectiveQA |
>     |-------|----------|---------------|-------------|
>     | Qwen3-14B (enable) | 73.07 | 91.30 | 72.73 |
>     | Qwen3-14B (disable) | 65.76 | 88.15 | 72.73 |
>     | Qwen3-8B (enable) | 67.01 | 84.63 | 69.48 |
>     | Qwen3-8B (disable) | 62.63 | 79.81 | 66.88 |
>
> ### Q2
> The intuition behind the performance drop for non-reasoning LLMs (Figure 3) is that reasoning tasks require procedural compatibility. As the number of CoT demonstrations increases, the prompt inevitably includes many examples that may look semantically similar but require entirely different solving strategies, like Table 1 in Appendix A.4.
>
> Furthermore, we would like to clarify that this decreasing and unstable trend is also observed in Figure 2 for reasoning tasks. In Figure 2, the cool-colored lines show instability and non-increasing performance as $n$ increases for non-reasoning models. To study this instability, we hope to explore the representative tasks. Therefore, we select geometry and number_theory, which show a relatively clear performance drop trend, to compare with reasoning LLMs.
>
> ### Q3
> Our fundamental setting explicitly includes CoT because we aim to explore how CoT demonstrations from in-domain tasks can guide the in-context learning of the LLM. "Ignoring the reasoning chain" contradicts the purpose of this setting, as it would mean the model falls back to relying almost entirely on its pretraining knowledge rather than learning from the provided in-context supervision.
>
> ### Q5
> Variations across datasets reflect the model's differing intrinsic capabilities, which we formalize as its "zone of understandable reasoning." Because baseline knowledge varies by domain, the threshold where models fail to smoothly internalize reasoning steps, causing instability occurs at different demonstration budgets.
>
> Regarding the accuracy drops, adding CoT demonstrations introduces instability for non-reasoning LLMs. Lacking explicit reasoning mechanisms (e.g., \<think\> tokens), these models struggle to extract cohesive procedural structures from long rationales, eventually becoming overwhelmed by accumulating noise and path-dependence.
>
> For wr > cr, we hypothesize flawed, self-generated trajectories (wr) might offer richer task understanding. Similar to test-time scaling, exploring incorrect paths exposes the model's inherent failure modes. This implicitly provides valuable intermediate structural information that can successfully pivot during inference.
>
> ### Q6
> Referring to Principle 1 (Ease of Understanding), self-generated CoTs are highly beneficial for weaker models because they align perfectly with the model's own generation distribution. However, as model capacity increases (e.g., moving from 8B to 14B), the model's ability to parse and internalize complex, external supervision also improves. This makes providing **origin** even better than **first** in DetectiveQA.

---

> > ### Author Rebuttal · Reviewer_X8ms · 2026-04-03
> >
> > The rebuttal provides helpful intuition, but many of the explanations remain largely qualitative and somewhat speculative. Additionally, the evaluation is somewhat limited in scope, both in terms of baseline comparisons and diversity of tasks. Therefore, I would like to keep my score.

---

> > > ### Author Response · Authors · 2026-04-06
> > >
> > > We sincerely thank you for your feedback. We would like to address your concerns regarding the qualitative nature of our explanations and the scope of our evaluation.
> > >
> > > 1. Broad and Representative Task Evaluation
> > >
> > >    To ensure the generalizability of our claims, we evaluate a wide range of tasks spanning both non-reasoning and reasoning domains (e.g., SuperGLUE, NLU, MATH,  DetectiveQA etc.) . By analyzing this broad spectrum, we located a critical gap in how scaling affects reasoning tasks specifically (a finding also recognized and appreciated by other reviewer). To provide an in-depth mechanistic analysis, we selected the most representative reasoning tasks to study the CoT dynamics thoroughly.
> > >
> > > 2. Comprehensive LLM Baselines Across Different Paradigms
> > >
> > >    We systematically evaluate across entirely different types of LLMs. This includes non-reasoning instruction-tuned LLMs (e.g., LLaMA and Qwen 2.5) and large reasoning LLMs (e.g., Qwen 3, QwQ-32B, DeepSeek-R1) . This distinct categorization is what allowed us to uncover that positive scaling in CoT-ICL strictly depends on the model's internal reasoning mechanisms.
> > >
> > > 3. Ablations to Validate Principles
> > >
> > >    To ensure our principles are empirically grounded rather than speculative, we conducted ablations. For instance, to validate the "ease of understanding" principle, we did not just compare self-generated CoTs to dataset-provided rationales, we also injected CoTs generated by stronger LLMs into the prompts of weaker LLMs . The results proved that distributional alignment (self-generation) is important for models to internalize the procedure, supporting our hypothesis. Additionally, we also conduct further ablation during rebuttal to strengthen our empirical findings.
> > >
> > > 4. A Cohesive, Quantitative Set of Evidence
> > >
> > >    Our findings are not isolated anomalies or qualitative guesses. They form a cohesive, mathematically quantified set of evidence that contradicts the traditional "pattern matching" hypothesis and points directly to an active, test-time learning process.
> > >    - The Failure of Semantic Similarity: If CoT-ICL were merely large-scale pattern matching, retrieving semantically similar examples should help. We quantitatively showed it actually harms reasoning performance.
> > >    - Growing Path Dependence (Order Sensitivity): The statistically measured variance (instability) for reasoning tasks grows as more CoT demonstrations are added, proving the model undergoes a path-dependent learning process.
> > >    - The Necessity of "Understanding" (Distributional Alignment): Models quantitatively perform better when prompted with their own self-generated rationales compared to dataset-provided gold rationales.
> > >    - The Need for Smooth Information Flow: We mathematically quantified "smoothness" by computing the trajectory curvature of demonstration embeddings.
> > >
> > > 5. Additional Experiments on Logic Reasoning (ProofWriter)
> > >
> > >    To further address your concern regarding task diversity, we have conducted new experiments on an additional domain: logic reasoning using the ProofWriter dataset, which provides explicit logical reasoning rules and validity claims. The results on ProofWriter perfectly align with our existing findings. Although our original empirical observations already showed robust consistency across math and narrative reasoning, this additional experiment further expands our evaluation scope and strengthens the validity of our proposed principles.
> > >
> > >     - Self-generate in-context CoT (Principle 1) with Qwen3-14B:
> > >
> > >         | | n=16 | n=32 | n=64 | n=128 |
> > >         | :--- | :--- | :--- | :--- | :--- |
> > >         | origin | 77.20 | 81.80 | 81.00 | 79.00 |
> > >         | first | 81.00 | 82.00 | 82.80 | 82.00 |
> > >
> > >
> > >     - CDS experiment with n=128 and bge-m3 (Principle 2):
> > >
> > >         || Qwen3-8B | Qwen3-14B |
> > >         | :--- | :--- | :--- |
> > >         | cds | 83.20 | 84.00|
> > >         | origin | 75.20 | 79.00 |
> > >
> > > We hope these clarifications and new experiments demonstrate the rigorous, quantitative nature of our work and address your concerns. Thank you again for your reviews, we will include more discussion and details in our revised manuscript.

---

### Official Review · Reviewer_KT4M · 2026-03-13

**Soundness:** 3
**Presentation:** 3
**Significance:** 3
**Originality:** 3
**Overall Recommendation:** 4
**Confidence:** 3

**Summary:**

This paper studies many-shot chain-of-thought in-context learning (CoT-ICL) and asks a fundamental question: when a long context is filled with many CoT demonstrations, how do models actually use them? Through a broad empirical study across non-reasoning and reasoning tasks, as well as standard instruction-tuned and reasoning-oriented LLMs, the paper shows that many-shot CoT behaves very differently from standard many-shot ICL: positive scaling with more demonstrations mainly appears in reasoning-oriented models, while similarity-based retrieval often fails in reasoning settings. Based on these findings, the authors propose to reinterpret many-shot CoT-ICL as a form of in-context test-time learning, and derive two principles for effective demonstration design: demonstrations should be easy for the target model to understand, and they should be arranged with smooth knowledge progression. To operationalize the latter, the paper introduces a curvature-based view of demonstration ordering, modeling demonstrations as a trajectory in embedding space, and proposes CDS (Curvilinear Demonstration Selection) to construct smoother sequences. Empirically, CDS improves performance over original ordering on several reasoning tasks, while the paper’s broader contribution is to offer a new conceptual lens for thinking about long-context reasoning prompts.

**Compliance With Llm Reviewing Policy:**

Affirmed.

**Final Justification:**

Overall, after two rounds of rebuttal, I believe the authors have succeeded in moving the paper from “primarily an interpretive narrative” toward “an empirical paper with substantially stronger support.” I am genuinely impressed by the authors’ diligence, persistence, and serious experimental attitude. After these two rounds of rebuttal, I believe they have significantly improved the quality of the paper and have come very close to resolving my fundamental concerns. Therefore, I have decided to change my score to WA.

**Key Questions For Authors:**

Please refer to the weakness part.

**Limitations:**

yes

**Strengths And Weaknesses:**

**Strengths**
 - **S1:** The paper is backed by substantial and carefully executed empirical work. Rather than presenting a single method with limited validation, the authors conduct a broad experimental study spanning non-reasoning vs. reasoning tasks, standard instruction-tuned vs. reasoning-oriented models, and multiple demonstration configurations. This makes the paper valuable not only for its proposed method, but also for the number of informative empirical observations it surfaces about how many-shot CoT-ICL actually behaves in practice.
 - **S2:** A major strength is that the paper identifies a real and important gap in the literature. Much of the existing intuition around many-shot ICL has been built on non-reasoning tasks, yet this work shows that these conclusions do not naturally transfer to the many-shot CoT reasoning setting. The paper asks a more fundamental question than a typical prompting paper: when long contexts are filled with many CoT demonstrations, how are those demonstrations actually used by the model? This is a meaningful question for both practice and theory, since it speaks directly to how reasoning-oriented LLM prompts should be designed.
 - **S3:** The paper also offers several insightful empirical findings. In particular, it shows that positive scaling with more CoT demonstrations appears mainly in reasoning-oriented models, whereas conventional instruction-tuned LLMs often do not benefit reliably and can even exhibit instability. It further shows that similarity-based retrieval, while effective in standard non-reasoning settings, breaks down on reasoning tasks, suggesting that semantic similarity is not an adequate proxy for procedural compatibility. These are useful and non-trivial observations, and they provide meaningful guidance for future prompt design in reasoning settings.
  - **S4:** Another notable strength is the originality of the smoothness/curvature perspective. Instead of treating demonstrations as an unordered set of examples, the paper models them as a trajectory in embedding space and uses curvature to quantify whether conceptual transitions between adjacent demonstrations are smooth. This is a creative shift in viewpoint: it reframes the problem from one of set selection to one of sequence geometry. Importantly, this idea is not presented only as intuition; the paper also provides empirical evidence that lower-curvature orderings correlate with better performance and uses this insight to motivate CDS. This is, in my view, one of the most distinctive and thought-provoking contributions of the paper.

---

**Weaknesses**
 - **W1:** The paper’s central conceptual claim appears somewhat stronger than the evidence provided. A major contribution of the paper is to reinterpret many-shot CoT-ICL as a form of in-context test-time learning, and to derive from this view two principles, namely ease of understanding and smoothness of knowledge progression. While this framing is intuitive and does help organize several empirical findings, the current evidence is still largely behavioral and indirect. The observed phenomena — such as stronger order sensitivity at larger shot counts, the failure of similarity-based retrieval on reasoning tasks, and the advantage of lower-curvature orderings — do suggest that demonstration ordering matters, but they do not yet directly establish that the model is internally “absorbing procedures” in a test-time learning sense. At present, this reads more as a plausible and insightful interpretive lens than as a mechanism that has been rigorously verified. Stronger support would require more direct mechanistic evidence, such as finer-grained analysis of intermediate states, controlled interventions on reasoning traces, or stronger causal ablations.
- **W2:** There is also a noticeable gap between the stated optimization objective and the actual solver used in CDS. The method is motivated as minimizing the total curvature of the demonstration trajectory, but in practice the paper does not optimize curvature directly. Instead, it constructs a graph in projected embedding space using Euclidean distances, then applies a TSP-based approximation to obtain a short path/tour and linearizes it into a final ordering. The concern is that a shortest path is not equivalent to a minimum-curvature path. Thus, there remains an under-justified jump between the conceptual objective (“smooth curriculum”) and the implemented optimization proxy (“short geometric route”). This approximation may be reasonable as an engineering heuristic, but the paper does not provide a sufficiently strong theoretical or empirical argument for why minimizing path length should reliably recover the desired low-curvature structure.
 - **W3:** The empirical evidence around curvature is suggestive, but still limited in scope and interpretability. The paper reports a negative correlation between curvature and downstream accuracy, including an overall correlation of r=−0.547, with similarly negative values on geometry, number theory, and counting/probability. These results are supportive, but several concerns remain. First, correlation is not causation: lower-curvature orderings may coincide with easier subsets of demonstrations or favorable local cluster structure, without demonstrating that “smooth progression” itself is the operative factor. Second, the curvature definition depends on a particular representation pipeline (Qwen3-Embedding-4B), including both the embedding model and the projection step. Third, the core curvature-performance analysis is conducted primarily on a small set of mathematical reasoning tasks. Although CDS is later tested on DetectiveQA, the broader claim that smoothness is a general principle for many-shot reasoning prompts is not yet validated across a wider range of reasoning settings.
 - **W4:** the representation dependence of the main claim is insufficiently examined. From the paper, the curvature analysis appears to rely on Qwen3-Embedding-4B as the embedding backbone, with PCA/UMAP-style projection variants applied afterward, but without showing whether the main findings are robust across alternative embedding models or embedding families. This raises an important question: are the reported curvature-performance trends capturing a fundamental property of demonstration ordering, or are they at least partly artifacts of a specific induced representation space? The paper does not currently show whether the negative curvature-performance relationship persists under different embeddings, whether CDS retains its advantage when the ordering is built from alternative representation backbones, or whether the “best” ordering is stable across embedding choices. Without such robustness checks, the broader claim that smoothness is a fundamental principle remains somewhat under-supported.

---

> ### Author Rebuttal · Authors · 2026-03-31
>
> We sincerely thank the reviewer for raising insightful reviews, we will address your concern in detail,
>
> ### Need of more direct mechanistic evidence:
> We agree that more direct evidence is needed to prove that the model is genuinely "absorbing procedural knowledge".
>
> To address this, we conducted an ablation study to isolate the absorption of procedural knowledge. For the reasoning-oriented models, we evaluated their performance at $n=128$ demonstrations (i.e., the context length that achieves near-optimal performance) with the below two settings:
>
> 1. Default (thinking enabled): The model processes the many-shot CoT context and generates intermediate reasoning (\<think\> tokens) before the final answer.
> 2. Ablation (thinking disabled): The model processes the identical many-shot CoT context but is forced to output the final answer directly, bypassing the generation of intermediate reasoning tokens.
>
> | Model | geometry | number_theory | DetectiveQA |
> |-------|----------|---------------|-------------|
> | Qwen3-14B (Default) | 73.07 | 91.30 | 72.73 |
> | Qwen3014B (Ablation) | 65.76 | 88.15 | 72.73 |
> | Qwen3-8B (Default) | 67.01 | 84.63 | 69.48 |
> | Qwen3-8B (Ablation) | 62.63 | 79.81 | 66.88 |
>
> With thinking disabled, models typically fail catastrophically. The context within the thinking token provides the procedural understanding to LLM. To further corroborate this, we analyzed the generated reasoning traces when thinking is enabled. For example, evaluating Qwen3-8B on geometry, increasing n from 16 to 128 improves accuracy from 66.18% to 73.07%. Crucially, alongside this performance gain, the average token length of the generated content within the \<think\>\</think\> tags decreased by 24.02%, showing the internalization of task understanding during the forward pass. Performing case studies on the think content also shows a higher confidence to answer in n=128.
>
> ### Concerns with CDS
> We agree that it left a gap between our stated objective (minimizing total curvature) and our implementation. To address this, we have conducted new experiments using the full embedding space with a combined distance metric that explicitly incorporates both Euclidean distance and heuristic local curvature.
>
> Relying solely on curvature (angles) can lead to paths that maintain a straight line but make massive leaps across the embedding space. This allows adjacent demonstrations to share some foundational context while avoiding sharp conceptual zig-zags.
>
> We have evaluated this combined approach. The computational cost remains highly practical: constructing the combined distance matrix and solving the TSP for the longest context evaluated in our paper $n \le 128$ completes in under minute on a standard CPU, which is negligible compared to the LLM inference time.
>
> Under this, we solidify our claims with the below settings,
> - We replaced the original embedding model with `bge-m3` to extract features and perform the CDS (denoted as bge-CDS), comparing it against the original setup (CDS).
> - To ensure the performance gains of CDS are not limited to the Qwen family, we evaluated an additional closed-source model, `gpt-5.2`.
>
> **number_theory**
> | Model | Method | n=16 | n=16 | n=16 | n=16 |
> | :--- | :--- | :--- | :--- | :--- | :--- |
> | gpt-5.2 | origin | 89.63 | 91.11 | 88.56 | 91.48 |
> | gpt-5.2 | CDS | 89.26 | 92.59 | 92.04 | 91.85 |
> | gpt-5.2 | bge-CDS | 88.15 | 91.48 | 88.70 | 91.30 |
> | Qwen3-14B | origin | 86.67 | 87.96 | 86.30 | 90.93 |
> | Qwen3-14B  | CDS  | 85.56 | 87.85 | 87.78 | 90.74 |
> | Qwen3-14B  | bge-CDS | 85.37 | 87.78 | 87.22 | 91.30 |
>
> **geometry**
> | Model | Method | n=16 | n=16 | n=16 | n=16 |
> | :--- | :--- | :--- | :--- | :--- | :--- |
> | gpt-5.2 | origin | 75.99 | 74.74 | 75.37 | 75.78 |
> | gpt-5.2 | CDS | 81.21 | 78.08 | 80.79 | 75.99 |
> | gpt-5.2 | bge-CDS | 80.37 | 78.29 | 76.83 | 75.11 |
> | Qwen3-14B | origin | 66.18 | 65.76 | 65.14 | 72.03 |
> | Qwen3-14B | CDS | 65.55 | 68.27 | 68.89 | 73.90 |
> | Qwen3-14B | bge-CDS | 66.60 | 67.85 | 70.36 | 74.32 |
>
>
> **DetectiveQA**
> | Model | Method | n=16 | n=16 | n=16 | n=16 |
> | :--- | :--- | :--- | :--- | :--- | :--- |
> | gpt-5.2 | origin | 80.52 | 82.47 | 83.77 | 85.71 |
> | gpt-5.2 | CDS | 80.52 | 83.12 | 85.06 | 88.31 |
> | gpt-5.2 | bge-CDS | 81.17 | 85.71 | 86.36 | 88.31 |
> | Qwen3-14B | origin | 75.97 | 74.03 | 70.78 | 72.73 |
> | Qwen3-14B | CDS | 76.62 | 75.32 | 73.38 | 75.32 |
> | Qwen3-14B | bge-CDS | 75.32 | 77.29 | 75.32 | 76.63 |
>
> Our evaluations demonstrate consistent performance growth across different embedding models and different target evaluation LLMs, confirming that the smoothness principle is a robust, representation-independent property of in-context learning. A plausible explanation for the marginal gain in number\_theory is that the evaluated LLMs already possess highly proficient, well-internalized knowledge for this domain, as evidenced by the high baseline accuracies. This leaves little room for demonstration ordering to yield further improvements.

---

> > ### Author Rebuttal · Reviewer_KT4M · 2026-04-03
> >
> > Reason: please refer to the comment I left after the authors' second rebuttal.

---

> > > ### Author Response · Authors · 2026-04-06
> > >
> > > We sincerely thank the reviewer for the detailed review. We will address your concern as follows,
> > >
> > > ### Direct Verification of Procedure-Absorption
> > > To directly verify the "procedure-absorption" mechanism, we conducted a "Procedural Corruption" ablation on the geometry task, which exhibits a strong positive scaling from $n=16$ to $n=128$, indicating a high potential for procedural absorption.
> > >
> > > We compared two settings using reasoning models:
> > > - *Valid Procedure:* Standard many-shot CoT with correct, problem-specific reasoning chains: $(x_i, C_i, y_i)$.
> > > - *Corrupted Procedure:* Questions and final answers are preserved, but all reasoning chains are replaced by the same static CoT (from the first demonstration): $(x_i, C_0, y_i)$.
> > >
> > > This controls perfectly for format, context length, and the $x \rightarrow y$ mapping, isolating the effect of specific procedural alignment. If models do not absorb procedures, both settings should scale similarly.
> > >
> > > | Model | Setting | $n=16$ | $n=128$ |
> > > | :--- | :--- | :--- | :--- |
> > > |Qwen3-8B | *Valid* | 57.62 | 67.01 |
> > > | | *Corrupted* | 57.62 | **65.76** |
> > > | Qwen3-14B | *Valid* | 66.17 | 73.07 |
> > > | | *Corrupted* | 67.01 | **70.56** |
> > >
> > > At a smaller shot count ($n=16$), the performance difference between the two settings is negligible, suggesting that the model relies more on its pre-trained priors or basic format matching based on query-level information, rather than actively learning from the provided procedure. However, at $n=128$, the Corrupted setting suffers a notable performance drop compared to the Valid setting.
> > >
> > > This provides direct mechanistic evidence for our claim. The degradation in the Corrupted setting at large $n$ proves that the model does not merely memorize $x \rightarrow y$ pairs or get passively "activated" by long contexts. Instead, it actively reads, extracts, and internalizes the specific procedural steps ($C_i$) provided in the demonstrations to refine its internal solution process during the forward pass when informative procedural information is provided.
> > >
> > >
> > > ### Causal Interpretation & Heuristic Optimizer
> > >
> > > To explicitly isolate the causal effect of a "smooth trajectory" (curvature) from "spatial proximity" (local clustering), we conducted a new controlled ablation study. Using the exact same set of demonstrations, we generated two different orderings by modifying the distance objective. Both orderings strictly constrain the Euclidean distance $D_{\text{euclidean}}$ to ensure the model traverses similar local cluster neighborhoods, but we inverted the curvature objective $D_{\text{curvature}}$ for the baseline:
> > >
> > > - `cds`: $D_{\text{cds}} = D_{\text{euclidean}} + D_{\text{curvature}}$
> > > - `high_curvature`: $D_{\text{high curv}} = D_{\text{euclidean}} + \left(\max(D_{\text{curvature}}) - D_{\text{curvature}} \right)$
> > >
> > > By doing this, the `high_curvature` ordering still groups semantically similar demonstrations together (preserving local clusters), but intentionally forces the TSP algorithm to take sharp, abrupt conceptual turns (maximizing the angle between consecutive steps). We further evaluated this with bge-m3:
> > >
> > > | Task | Model | Strategy | n=16 | n=32 | n=64 | n=128 |
> > > | :--- | :--- | :--- | :--- | :--- | :--- | :--- |
> > > | number_theory | gpt-5.2 | cds | 88.15 | 91.48 | 88.70 | 91.30 |
> > > | | | high curv| 85.74 | 89.63 | 86.48 | 88.15 |
> > > | | Qwen3-14B | cds | 85.37 | 87.78 | 87.22 | 91.30 |
> > > | | | high curv | 79.26 | 84.44 | 84.26 | 90.37 |
> > > | geometry | gpt-5.2 | cds | 80.37 | 78.29 | 76.83 | 75.11 |
> > > | | | high curv | 72.65 | 73.90 | 76.33 | 74.53 |
> > > | | Qwen3-14B | cds | 66.60 | 67.85 | 70.36 | 74.32 |
> > > | | | high curv | 66.38 | 66.81 | 66.60 | 71.80 |
> > >
> > > The `cds` strategy consistently outperforms `high_curvature`. Since both orderings control for local cluster density and use identical demonstrations, this isolates abrupt transition angles as the causal factor hindering performance. This proves ICL fundamentally benefits from a smooth pedagogical progression, not just from grouping similar items.
> > >
> > > Regarding the observation that CDS is not a direct optimizer of the stated objective, we chose a heuristic approach due to computational tractability. Exactly minimizing curvature for n demonstrations requires evaluating $n!$ orderings. For n=128, the search space ($128!$) makes exact optimization intractable for on-the-fly prompt construction.
> > >
> > > Our core theoretical claim is that a 'smooth progression' (i.e., sufficiently low curvature) facilitates in-context test-time learning, and it does not strictly require the absolute mathematical global minimum. Empirically, the TSP-based heuristic successfully reduces trajectory curvature and yields significant performance gains. These results validate that our heuristic approximation is practically sufficient to achieve the benefits of smooth information flow.
> > >
> > > We hope these address your concerns, and we will include all the details in the revised manuscript. Thank you for your detailed reviews that help us strengthen our work.

---

### Decision · Program_Chairs · 2026-04-30

**Decision:**

Accept (regular)

**Comment:**

This paper investigates the scaling behaviors of many-shot CoT in-context learning, proposing that models engage in "in-context test-time learning" rather than simple pattern matching, and introduces Curvilinear Demonstration Selection (CDS) to optimize demonstration ordering. Initially, reviewers appreciated the counter-intuitive findings but raised significant concerns regarding the lack of direct mechanistic evidence, statistical variance, and representation dependence. In response, the authors provided causal evidence and multi-seed variance analyses via additional experiments and largely convinced the majority of the reviewers. The AC took a look a the paper and agree with the majority reviewers' assessment on the value of the paper and would like to (weakly) recommend acceptance,

Having said that I do wish to bring up a missing citation [1] which explicitly documents the several core phenomena this current submission claims as novel findings— e.g., that scaling many-shot ICL can lead to performance plateaus or degradation, 2) that performance is often disproportionately tied to a small subset of influential examples and 3) similarity-based retrieval in long-context ICL is often ineffective. The authors should properly discuss this prior work in the camera ready paper.

[1] From Few to Many: Self-Improving Many-Shot Reasoners Through Iterative Optimization and Generation. ICLR 2025.